

# Different tracer, different bias: using radon to reveal flow paths beyond the Window of Detection

Mortimer L. Bacher[1,2], Julian Klaus[2], Adam S. Ward[3], Jasmine Krause[3], Catalina Segura[4], Clarissa Glaser[2]

[1] Department of Geosciences, University of Tübingen, Tübingen, Germany
[2] Department of Geography, University of Bonn, Bonn, Germany
[3] Department of Biological and Ecological Engineering, Oregon State University, Corvallis, OR, USA
[4] Forest Engineering, Resources, and Management, Oregon State University, Corvallis, OR, USA

*Correspondence to*: Clarissa Glaser (cglaser@uni-bonn.de)

**Abstract.** Slug tracer experiments have greatly advanced our understanding of solute transport in streams. Breakthrough curves (BTCs) from these experiments are biased toward faster flow paths, highlighting the need for alternative tracers to cover longer timescales. The radioactive tracer radon ($^{222}$Rn) is increasingly used to quantify transit times in subsurface transient storage zones, as it traces transit times of up to 21 days. However, it remains unclear whether calibrating transient storage models (TSMs) with radon yields longer subsurface timescales of transit times - and thus greater transient storage areas - than with slug tracers such as sodium chloride (NaCl). To address this, we conducted radon measurements and NaCl slug tracer experiments in Oak Creek (Oregon, USA) and jointly and individually calibrated TSM parameters with both tracers. We applied parameter identifiability analysis and evaluated the information provided by both tracers in constraining model parameters. Our results show that calibrating the TSM with radon and chloride increases information on model parameters compared to calibrating the TSM with each tracer individually. This suggests that incorporating radon in calibration improves solute transport estimates in future studies. However, when calibrating the TSM with only radon measurements, all resulting parameters of the TSM were non-identifiable. This non-identifiability arises from steady state activity of radon in streams and radon's high sensitivity to the amount and location of groundwater inflow, which is not explicitly accounted for in TSMs. As a result, radon measurements are biased toward longer-timescale flow paths, limiting its usefulness for characterizing solute transport in calibrating TSMs without chloride.

## 1 Introduction

The time a water parcel spends in river corridors is a key variable controlling biogeochemical processes and the ecological functioning of streams (Harvey and Gooseff, 2015; Ward and Packman, 2019). In environmental systems, the ensemble of these timescales for many parcels of water are termed transit time distributions (TTDs). Empirical studies of TTDs in streams typically rely on solute tracer experiments (e.g., Stream Solute Workshop, 1990), which involve releasing a known mass of solute tracer into the stream and measuring its concentration over time, i.e., the breakthrough curve (BTC), at a downstream





location (Day, 1976). Despite their widespread use, solute tracer experiments are biased toward measuring faster flow paths within TTDs due to the 'window of detection' (WoD). The WoD refers to the longest temporal scale of tracer-labelled flow paths that contribute to measurable tracer concentrations distinguishable from the background concentration (Harvey et al. 1996; Wagner and Harvey 1997; Ward et al. 2023), ultimately defining the longest timescales that can be observed in a given study. Despite decades of research (Harvey and Bencala, 1993; Wagner and Harvey, 1997), measuring flow paths with timescales beyond the WoD remains a challenge, leaving critical gaps in our understanding of solute transport in streams. This underscores the need for new approaches to capture these overlooked timescales.

Solute tracer studies are often evaluated by calibrating transient storage models (TSMs) to match empirical BTCs. TSMs assume a uniform, steady-state, one-dimensional flow, modeled using the advection-dispersion equation (ADE), while also accounting for first-order mass transfer between the advective flow and a storage zone of effectively infinite dimensions (Bencala and Walters, 1983; Gooseff et al., 2008). Water in the storage zone is delayed relative to the main channel flow and is located in surface transient storage zones within the channel (Nordin and Troutman, 1980), either due to eddies and turbulence caused by in-stream obstructions (Jackson et al., 2013) or in subsurface transient storage zones (i.e., the hyporheic zone; Bencala and Walters, 1983; Cardenas and Wilson, 2007). The parameter values derived from TSMs provide a means of comparing solute transport within a single stream or across multiple streams. Despite the widespread use of TSMs, their application often produces contradictory results (Ward and Packman, 2019) due to two fundamental issues. First, model parameters are frequently non-identifiable, meaning that multiple parameter combinations can yield equivalent model performance. One approach to addressing this issue has been the application of parameter identifiability analysis. Previous studies highlighted the importance of incorporating identifiability analysis when calibrating TSMs with BTCs to enhance certainty of model parameters (Bonanno et al., 2022; Camacho and González, 2008; Kelleher et al., 2013; Wagner and Harvey, 1997; Wagener et al., 2002). In addition to identifiability analysis, adding observations is another commonly used strategy for reducing parameter uncertainty and improving model constraints (e.g., Nearing and Gupta, 2015). Research has demonstrated that incorporating additional tracer observations in TSM applications enhances the accuracy of solute transport estimations (Briggs et al., 2009; Neilson et al., 2010a; 2010b).

The second fundamental issue leading to contradictory results from TSMs is that they can only fit observed solute tracer data, meaning they do not account for flow beyond the WoD. This limitation is critical, as a growing body of research highlights the presence of flow paths that exceed the duration of slug tracer experiments (e.g., Ward et al., 2023). Specifically, tracer mass that is released but remains unrecovered (i.e., 'lost') within the WoD may either follow flow paths that exceed the duration of slug tracer experiments or bypass the downstream sampling location entirely by traveling through subsurface pathways (e.g., Covino et al., 2007; Payn et al., 2009). These subsurface flow paths can occur at multiple scales, exhibiting a wide range of transport times and distances (Cardenas, 2008). They play a crucial role in buffering temperature signals before returning to the channel (Briggs et al., 2022; Wu et al., 2020) and serve as reservoirs of exchange for shorter hyporheic flow paths that may mix with this water before re-entering the main channel (Payn et al., 2009). Some studies suggest that adapting study designs can effectively trace the entire continuum of subsurface flow paths, including large-scale exchange along the river



corridor (Covino et al., 2011; Mallard et al., 2014; Ward et al., 2023). However, measuring flow paths beyond the WoD at the reach scale remains challenging calibrating TSMs with 'traditional' measurements of solute tracer concentration.

The naturally occurring radon ($^{222}$Rn) may present an opportunity as a tracer to enhance our measurements of flow paths longer than the duration of slug tracer experiments. Radon has frequently been used to estimate transit times in subsurface transient storage zones (Cranswick et al., 2014; Frei et al., 2019; Gilfedder et al., 2019; Lamontagne and Cook, 2006; Pittroff et al., 2016) and to quantify groundwater inflows into streams (Cook et al., 2006; Cook, 2013). Radon is a radioactive noble gas that is produced through the decay of radium-226 ($^{226}$Ra), a parent isotope found in radium-bearing minerals in streambeds (Sakoda et al., 2011). As $^{226}$Ra decays, radon activity increases exponentially until secular equilibrium, which occurs when radon production equals its decay. This equilibrium also defines the maximum achievable radon activity based on the availability of radium-bearing minerals. For radon, secular equilibrium is established after approximately 21 days, which is about five times its half-life of 3.18 days (Krishnaswami et al., 1982). Secular equilibrium is maintained in aquifers because they are typically closed systems, preventing radon from readily escaping into the atmosphere. Radon activity in groundwater can exceed 100,000 Bq m$^{-3}$ (Cecil and Green, 2000), whereas surface water activity is usually several orders of magnitude lower due to atmospheric degassing. When surface water exchanges with subsurface transient storage zones and contacts radium-bearing minerals in the streambed, radon activity increases as a function of the time spent in the hyporheic zone. As a result, radon activity in streams offers insights into the duration that water parcels remain in the subsurface (i.e., in contact with radium-bearing minerals), particularly for transit times of less than 21 days. Subsurface transit times of up to 21 days exceed those measured in slug tracer experiments, where transit times usually range from minutes to hours.

The overarching goal of this study is to quantify flow paths of different timescales at the reach scale using measurements of solute tracer and naturally occurring radon. We expect that calibrating the TSM with radon and determining the model parameters will result in longer timescales of flow paths and, in turn, larger transient storage areas compared to calibration with 'traditional' slug tracer data. To test this expectation, we address the following questions:

● How do the values of model parameters and their identifiably differ when calibrating a TSM with only radon or chloride for the same study reach?

● How do parametric values and parameter identifiably change when jointly calibrating a TSM with radon and chloride, compared to calibrating each tracer individually?

To answer these questions, we apply a coherent mathematical framework to radon and slug tracer injections of sodium chloride (NaCl). Applying a coherent framework was motivated by findings from previous catchment-scale studies, which showed that applying the same models to different tracers yields a similar understanding of hydrological transport processes (e.g., Rodriguez et al., 2021; Wang et al., 2023). To ensure a coherent mathematical framework, we adapt the transient storage model OTIS ('One-Dimensional Transport with Inflow and Storage model,' Runkel, 1998) by incorporating radon-specific processes (e.g., degassing). We then jointly calibrate this model with slug tracer data (chloride) and with radon. We apply a global



analysis approach to ensure the identifiability of model parameters in our calibration approach. Subsequently, we use information theory to calculate the information gain from the joint and individual calibration of the TSM with these tracers.

## 2 Materials and Methods

### 2.1 Field site and experiments

We carried out slug tracer experiments and radon measurements in August 2023 in a 578-m long segment of Oak Creek (44°36'29.16"N, 123°19'56.05"W) in Oregon (USA) (Cargill et al., 2021; Katz et al., 2018; Milhous, 1973). Oak Creek mainly features basaltic lithology and stream sediments consist of cobble to gravel-sized weathering products of these basalts. We subdivided the selected Oak Creek segment into five reaches with lengths between 67 and 140 m (Fig. 1). Each reach length was at least 20 times the Wetted Channel Width to control for expected variations in solute transport that occur as a function

of reach selection (Anderson et al., 2005; Becker et al., 2023; Day, 1977). We equipped the upstream and downstream location of each reach with conductivity loggers (CTD Diver from Eijkelkamp Soil & Water and Levelogger of Solinst, both with an accuracy of ±1% for conductivity, ±0.1°C for temperature, and ±0.5 cm for the water pressure). We prepared NaCl solutions using one gallon (3.8 L) of stream water and studied each reach by conducting a slug tracer injection both upstream and downstream. The amount of NaCl varied between 500 and 2500 g and was adapted for each injection location to ensure that

the tracer signal at the reach's downstream location was elevated by at least 50 µS cm$^{-1}$ above the background concentration. We determined the injection locations to ensure lateral and vertical mixing of stream water with the injected solution of solute tracers by the time the tracer entered each study reach (Payn et al., 2009; Ward et al., 2013). After tracer injection, we measured electrical conductivity (EC) every 5 seconds and normalized it to 25°C. We then corrected for background EC of the stream and corrected EC to chloride concentrations based on EC-concentration regression lines (R² = 0.99). Discharge was calculated

from the resulting BTCs using dilution gauging (Kilpatrick and Cobb, 1985). Additionally, we determined the mass recovery of the injected tracer from the BTCs of the upstream and the downstream injection to quantify the amount of NaCl tracer mass lost during the experiment (Payn et al., 2009).




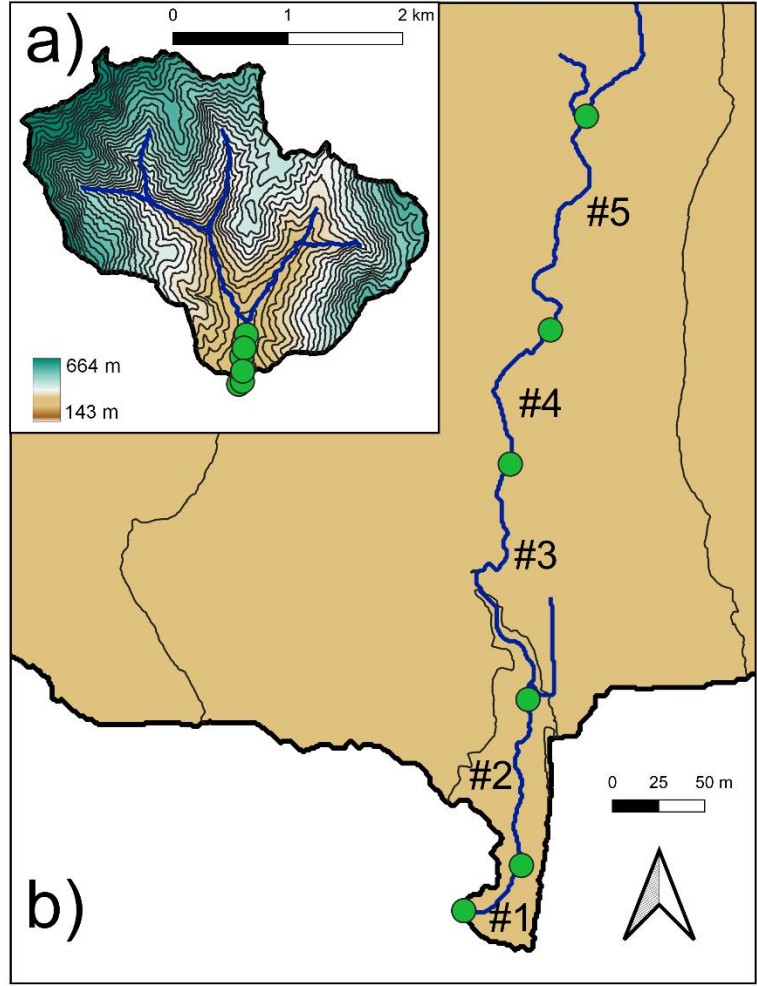

**Figure 1: a) Map of the Oak Creek catchment, with colors indicating elevation. Green markers denote tracer measurement locations. (b) Close-up view, where labels (#1–#5) represent reaches between these measurement locations.**

Radon sampling sites were co-located with BTCs observations, collected one day before the slug tracer injections. We filled one-liter amber glass bottles with stream water in the thalweg and tightly closed the bottles beneath the water surface to prevent degassing during sampling. The water samples were analyzed using a large dome, large detector RAD7 device (Durridge Company, Inc.). We employed a closed air loop approach as outlined by Lee and Kim (2006). Each one-liter water sample underwent degassing with the integrated pump of the RAD7 to strip the dissolved radon into the air phase. Subsequently, radon in the air of the closed-loop was counted up to six times over 30 minutes with the integrated detector of the RAD7. The resulting average value of the repeated radon measurements was corrected for the time between sampling and measurement to account for radioactive decay. Radon measurements were then multiplied by an empirical correction factor to adjust for differences in degassing between one-liter samples and the reference volume of the RAD H$_2$O method (250 mL).





We determined the maximum radon activity that can be achieved based on the available radium-bearing minerals of Oak Creek (i.e., the radon activity reached at secular equilibrium). For this purpose, we selected two locations at Oak Creek and collected five sediment samples from each location. Subsequently, these sediment samples were merged into two bulk samples to reduce

the potential for small-scale spatial heterogeneity in radon activity in the stream sediment, and to ensure representative sediment samples. We then conducted incubation experiments with stream sediment and radon-free water (after Corbett et al. 1997; 1998; Peel et al., 2022). After incubating the bulk sediment samples for at least 21 days inside gas-tight two-liter containers, we analyzed the water from these samples using the closed-loop approach with the RAD7, as described previously for the surface water samples. We assumed the same mineralogical composition of the aquifer and the streambed sediment.

This means the radon activity at secular equilibrium represents both the radon activity of the groundwater and the highest achievable activity along subsurface flow paths.

## 2.2 Transient storage modelling

Solute transport is commonly derived from calibrating the TSM to BTCs. The TSM describes the combined effect of flow velocity and dispersion on solute transport in a one-dimensional steady flow domain (Taylor 1922; 1954), dilution (or enrichment) of solutes from lateral groundwater inflow, and additionally considers a first-order mass exchange of solutes between the surface and a finite-size, well-mixed transient storage zone. The partial differential equations of the TSM are (Bencala and Walters 1983):

$$\frac{\partial C}{\partial t} = -\frac{Q}{A}\frac{\partial C}{\partial x} + \frac{1}{A}\frac{\partial}{\partial x}\left(AD\frac{\partial C}{\partial x}\right) + \frac{q_I}{A}(C_I - C) + \alpha(C_{TS} - C),$$

$$\frac{\partial C_{TS}}{\partial t} = -\alpha\frac{A}{A_{TS}}(C_{TS} - C)$$

*(1)*

where $C$ is the observed tracer concentration above the background concentration [M L$^{-3}$], $t$ is time [T], $Q$ the discharge in the stream channel [L$^3$ T$^{-1}$], $A$ the channel's cross-sectional area [m²], $x$ the distance [L], $D$ the longitudinal dispersion coefficient

[L² T$^{-1}$], $q_I$ the groundwater inflow into the stream channel [L$^3$ L$^{-1}$ T$^{-1}$], $\alpha$ the transient storage exchange coefficient [T$^{-1}$], $C_{TS}$ the solute concentration in the transient storage zone [M L$^{-3}$], and $A_{TS}$ the cross-sectional area of the transient storage zone [L$^{-2}$]. In case of a lack of exchange where $\alpha = 0$, eq. 1 reduces to the advection-dispersion equation.

The model formulation above is not suited for radon activity, as radon activity changes due to radioactive decay, degassing, and production in the transient storage zone (Cook 2013; Frei and Gilfedder 2015). Therefore, we implemented additional

radon specific processes in the one-zone TSM as follows:



$$\frac{\partial C}{\partial t} = -\frac{Q}{A}\frac{\partial C}{\partial x} + \frac{1}{A}\frac{\partial}{\partial x}\left(AD\frac{\partial C}{\partial x}\right) + \frac{q_I}{A}(C_I - C) - \lambda C - \frac{k}{d}C + \alpha(C_{TS} - C)$$

$$\frac{\partial C_{TS}}{\partial t} = -\alpha\frac{A}{A_{TS}}(C_{TS} - C) - \lambda C_{TS} + \gamma,$$

*(2)*

where $\lambda$ [T$^{-1}$] is the radioactive decay rate [T$^{-1}$; 0.18 d$^{-1}$ for radon], $k$ [L T$^{-1}$] the gas exchange velocity, $d$ [L] the stream depth, and $\gamma$ [M L$^{-3}$ T$^{-1}$] the production of radon in the transient storage zone. In the absence of radon-specific processes ($k = 0$, $\lambda = 0$ and $\gamma = 0$), eq. 2 reduces to the TSM described in eq. 1.

## 2.3 Numerical implementation of radon-specific processes in OTIS

The 'One Dimensional Transport with Inflow and Storage' (OTIS) model (Runkel, 1998) is one of the most commonly used implementations of the TSM. OTIS uses a Crank-Nicolson numerical scheme to solve the TSM. We adapted the existing code of OTIS (written in FORTRAN 77) to simulate radon activity. Hereafter, we will refer to the implementation of the TSM that considers radon-specific processes as OTIS-R (R for radon; Bacher et al., 2025).

## 2.4 Model calibration

We used measured chloride concentrations and radon activity to calibrate the model parameters of the TSM. The calibration was done in a Monte Carlo approach to assess the model performance for different combinations of parameter values (after Kelleher et al, 2013; Ward et al, 2017; 2018). We refer to a single combination of calibrated parameter values as a 'parameter set.' The same parameter sets were tested for both tracers to evaluate model performance for each tracer. The model performance was evaluated using the normalized root mean squared error (nRMSE). We performed this normalization to
enable a relative comparison of both tracers. We conducted three different calibration approaches, each with 200,000 iterations. The model parameters were sampled using Latin Hypercube Sampling (LHS), a method that employs stratified sampling while retaining the simplicity and objectivity of fully random sampling (Helton and Davis, 2003). In all three calibration approaches, we calibrated $D$, $\alpha$, and $A_{TS}$ by sampling them from a defined parameter range (Table 1). We assumed a uniform parameter distribution for $D$, $\alpha$, and $A_{TS}$, sampling from a log10 transformed distribution to ensure approximately equal representation
for each order of magnitude within the parameter space (Kelleher et al. 2013; Ward et al. 2017). We extracted the 1% and 10% with the lowest values for the nRMSE of the parameter sets and considered them as behavioral parameters (Beven and Binley, 1992). We selected these behavioral thresholds to ensure consistency with previous solute transport studies (e.g., Bonanno et al., 2022; Kelleher et al., 2019; Wagener et al., 2002; Ward et al., 2013, 2017; Wlostowski et al., 2013). Since we tested the same combinations of parameter values in the TSM for both tracers, the intersection of the behavioral parameter sets from both
tracers reflects the parameter sets obtained when the model is calibrated with both tracers together. This indicates that when





the behavioral parameter sets for both tracers are the same, the choice of tracer does not impact the parametric information related to solute transport. Still, other parameter sets were in the behavioral set for only one tracer but not the other, representing parameters with acceptable performance for a single tracer but not robust in describing both tracers. The behavioral parameter sets were used for all further analysis and calculations.


**Table 1: Model parameters used in OTIS-R. Parameter ranges are shown for those that were calibrated.**

| Model parameter | Range |
| --- | --- |
| $D$ [m² s⁻¹] | 1e-5 to 10 |
| $A$ [m²] | 1e-5 to 0.1 |
| $A_{TS}$ [m²] | 1e-5 to 100 |

Model parameters other than those calibrated - including the stream velocity ($v$), the cross-sectional area ($A$), the production term of radon in the storage zone ($\gamma$), and the gas exchange velocity ($k$) - were calculated before calibration. This reduces

potential issues of equifinality with TSMs (Knapp and Kelleher 2020). We calculated $v$ by dividing the stream length by the arrival time of the concentration peak of the downstream BTC, and calculated $A$ from these two parameters after the calibration approach. This choice was motivated by findings from Bonanno et al. (2022), who showed that $A_{TS}$ and $\alpha$ are often not identifiable when $v$ is calibrated instead of calculating v by dividing the stream length by the arrival time of the concentration peak of the downstream BTC. We calculated the radon production term $\gamma$ as the product of the decay constant (0.18 d⁻¹) and

the measured equilibrium radon activity (Gilfedder et al. 2019). For the gas exchange velocity, we relied on gas tracer experiments previously conducted at the same stream section as our study at Oak Creek (Cargill et al. 2021). We scaled the gas exchange coefficients for $SF_6$ reported by Cargill et al. (2021) to radon (Jähne et al., 1987; Raymond et al., 2012). We tested two different gas exchange velocities for each run of our three calibration approaches to quantify the uncertainty of degassing in calibration of the TSM. The gas tracer experiments by Cargill et al. (2021) were conducted at three different

discharge conditions (0.05 m³s⁻¹, 0.1 m³s⁻¹, 1.07 m³s⁻¹). We used values from the experiments conducted during the lowest and highest discharges for parameterization. This resulted in one model setup with a low gas exchange value and another with a high gas exchange value. Hereafter, we will refer to these different values used for parameterization as $k_{low}$ ($k_{600}$ = 206 d⁻¹) and $k_{high}$ ($k_{600}$ = 290 d⁻¹).

**2.5 Evaluating parameter sensitivity, certainty, and interactions**

After model calibration, we evaluated the parameter identifiability of the behavioral parameters through sensitivity, certainty, and interactions analysis. We refer to a parameter set as 'identifiable' when the values of the model parameters are certain, sensitive, and do not have any parameter interactions. These parameter identifiability analyses include the visual inspection of





I) nRMSE vs. parameter plots (Wagener et al., 2003), II) cumulative parameter distribution plots (Kelleher et al., 2019), III)
posterior distribution plots (Wagener et al., 2002; Ward et al., 2017), and IV) scatter plots of the behavioral parameters. In the
nRMSE vs. parameter plots, parameters needed to exhibit a distinct peak of performance in nRMSE vs. parameter plots to be
categorized as sensitive and certain. The cumulative distribution functions (CDFs) of the top 1% or 10% of results (behavioral
parameters) had to visibly deviate from the 1:1 line (representing a uniform distribution) to be categorized as sensitive. The
probability density functions (posterior distributions) had to be peaked and narrow to categorize parameters as certain.

The posterior distribution of parameters is represented by the histogram of behavioral parameter sets and their performance
(nRMSE). This histogram was created by dividing the parameter values into 15 equally sized bins, where the bar height
illustrates the likelihood of a parameter falling within a specific bin. Sensitive parameters exhibit higher variation in likelihood
across different parameter values. In the scatter plots, narrow and constrained values of two model parameters indicate
identifiable parameters. Parameter interactions are visible through changes in one parameter's value relative to changes in
another within the parameter space. These interactions are visually depicted as a curve in the parameter space, suggesting that
variations in parameter 1 result in corresponding variations in parameter 2.

For a quantitative measure of the parameter sensitivity that underpins the visual inspections, we applied the two-sample
Kolmogorov–Smirnov (K-S) test that calculates the maximum distance $K$ and the corresponding *p-value* between two
cumulative distribution functions:

$$[K, p] = max \, |F(P_{behavioral}) - F(P_{non-behavioral})|$$

*(3)*

where $F(P_{behavioural})$ and $F(P_{nonbehavioral})$ are the cumulative distribution functions of a parameter P for the behavioral and
non-behavioral parameter sets, respectively. The K-S test thus expresses the degree of sensitivity of a parameter. We grouped
parameter sensitivity into four different categories following the approach of Ouyang et al. (2014): highly identifiable (K >
*0.2*; *p-value* ≤ 0.05), moderately identifiable 0.1 ≤ K ≤ 0.2; *p-value* ≤ 0.05), poorly identifiable *(K < 0.1; p-value ≤ 0.05)* and
non-identifiable *(p-value > 0.05)*. Moreover, we calculated Spearman rank correlation coefficients ($p_{spearman}$) between different
model parameters for a quantitative measure of the visual inspection of the scatter plots of model parameters. Given the non-
linear nature of these interactions, we used the non-parametric Spearman rank correlation coefficient, with a *p-value* of 0.05
for determining statistical significance.



### 2.6 Evaluating information content of model parameters

We used the information content to evaluate the certainty of the model parameters by calculating the Shannon entropy of the posterior parameter distributions of these parameters (Cover and Thomas, 2005; Loritz et al., 2018). The posterior parameter
distribution is the probability density function of the behavioral parameters sets. The Shannon entropy reads:

$$H(X|T) = -\sum_{k=1}^{n_I} f(I_k) log_2 f(I_k)$$

*(4)*

Where $H$ describes the Shannon entropy and $X$ the parameter of interest. $T$ is the tracer, for which behavioral parameter sets
were extracted. The tracer could either be radon, chloride or a combination of both ($H(X|(radon \cap chloride))$). We binned the parameter values into 15 bins of equal size, similar to visual inspection of the posterior distribution of the parameter certainty. The rationale for choosing 15 bins was that the resulting histograms visually revealed the underlying structure of the parameter values without introducing uneven features, such as spiky histograms. The height of each bin describes the likelihood of the parameter being located in this specific bin. $n_I$ [-] signifies the number of intervals (bins), and $f(I_k)$ [-] describes the probability
of the parameter $X$ falling within the interval $I_k$. $f(I_k)$ describes the probability of the parameter $X$ to take a value in an interval $I_k$ for radon or chloride, a combination of both (posterior distribution), or none of those (prior distribution). Smaller values of $H$ show that the distribution is not flat and that it is more certain than a uniform prior distribution.

Furthermore, we evaluated the information gain from the prior to the posterior distribution of model parameters when the TSM was calibrated with radon and chloride separately, as well as for both tracers together. In this context, the prior distribution
describes the uniform distribution of all parameters prior to parameter calibration. The minimum and maximum values for this distribution are defined through the parameter range from which these parameters were sampled (Table 1). The information gain quantifies how much information the tracers add to the model parameters when calibration of the model was conducted with these tracers separately and with both tracers together. We evaluated the information gain from prior to posterior distributions for each model parameter using the Kullback-Leibler divergence $D_{KL}$ (Rodriguez 2021):


$$D_{KL}(X|T,X) = \sum_{k=1}^{n_I} f(I_k) log_2 \frac{f(I_k)}{g(I_k)}$$

*(5)*

where $g(I_k)$ [-] is the probability of the parameter $X$ to fall in the interval $I_k$ in the prior distribution. Higher values of $D_{KL}$ (in
bits) show a higher information gain from prior to posterior parameter distribution during calibrating the model (Rodriguez



2021). Summing values of the Kullback-Leibler divergence for all TSM parameters yields the total information on solute transport from that tracer.

## 2.7 Considering groundwater inflow for calibrating the TSM

Radon activity in streams varies with the amount of inflowing groundwater, as radon activity differs significantly between groundwater and surface water (Cook, 2013). Small changes in the amount of inflowing groundwater may lead to differences in model performance. To account for this, we either calibrated or calculated groundwater inflow within three different calibration approaches, in addition to calibrating $D$, $\alpha$, and $A_{TS}$. These approaches to handling groundwater inflow were selected to assess how varying values of groundwater inflows affect model performance. For the first calibration approach ($Q_{fix}$), we

calculated the groundwater inflow by dividing the difference of discharge between the upstream and downstream BTCs by the reach length (Table 1). In the second calibration approach, we calibrated discharge ($Q_{LHS}$) and subsequently calculated groundwater inflow. Discharge was sampled from a normal distribution, because Schmadel et al. (2010) reported that discharge measurement errors follow a normal distribution. We used the calculated discharge as the mean of the normal distribution and assumed that its standard deviation represents the uncertainty of dilution gauging. In the third calibration approach ($Q_{out}$), we

calculated groundwater inflow from the calibrated discharge. This direct calibration of groundwater inflow allows us to calculate gross water fluxes along reaches. This is because OTIS, and by extension OTIS-R, accounts for water mass balance by parameterizing groundwater inflow using the following equation:

$$\frac{\partial Q}{\partial x} = q_I\text{-}q_{out}$$

300                                                                                *(6)*

where $q_I$ [L² T⁻¹] is the gross water inflow and $q_{out}$ [L² T⁻¹] the gross water outflow from the stream into the groundwater. In the first and second calibration approaches, gross water outflow was assumed to be zero. These gross water fluxes are commonly derived from calculating the mass loss of BTCs relative to the injected tracer mass (i.e., 'channel water balance'; Payn et al., 2009).

Additionally, radon activity in streams depends on the location of the groundwater flow (Cook 2013). Assuming groundwater inflow as either discrete or linear in the TSM might therefore lead to different simulated radon activity and, in turn, to different calibrated parameter values to achieve a good fit between simulated and measured radon activity. To test this, we calibrated the TSM with radon activity and chloride concentrations for the downstream-most reach in three different model setups. In each setup, we assumed different locations of groundwater inflow along the selected reach (upstream-most point, mid-point,

and downstream-most point in the study reach). The sub-reach with the groundwater inflow attributed to a 1 m long subdivision and at a magnitude equal to the total net increase in stream flow along the study reach (i.e., all groundwater inflow in a single



location, representing a focused discharge of groundwater into the stream, like that attributable to a geologic discontinuity or fracture). For each of the three model setups, we ran the model 200,000 times and calibrated the model parameters ($D$, $\alpha$, $A_{TS}$, discharge) following the same procedures as the prior model fitting. We then compared calibrated model parameters among the different model setups and applied identifiability analysis to the behavioral parameter sets (1% and 10%). Subsequently, we applied the Levene test for equality of variance to compare the distributions of the model parameters from the different model setups (upstream, middle, downstream), using a *p-value* of 0.05 for determining statistical significance.

## 3 Results

### 3.1 Tracer concentration in Oak Creek

BTCs (Fig. S1) and measured radon activity revealed spatial variability across the study reaches. Radon activity in the stream ranged from 285 ($\pm$ 22) Bq m$^{-3}$ to 337 ($\pm$ 26) Bq m$^{-3}$. The highest activity was observed at reach #2. Radon activity in groundwater was 23 times higher than in surface water, reaching 6765 ($\pm$ 841) Bq m$^{-3}$.

### 3.2 Information content and information gain for model parameters

Calibrating the TSM with both tracers resulted in higher values of the Kullback-Leibler divergence and thus more information on model parameters compared to calibration with chloride or radon individually ( $D_{KL}(X|(radon \cap chloride),X) > D_{KL}(X|chloride,X)$ and $D_{KL}(X|(radon \cap chloride),X) > D_{KL}(X|radon,X)$ ) (Table 2). When calibrating the TSM with each tracer individually, calibration with chloride yielded more information on model parameters than calibrating with radon ($D_{KL}(X|chloride,X) > D_{KL}(X|radon,X)$). The only exception occurred in the calibration approach with fixed groundwater inflow ($Q_{fix}$). Values of the Kullback-Leibler divergence of individual parameters varied depending on the tracer used for calibration. In general, chloride provided more information on dispersion but little information on groundwater inflow, whereas radon provided more information on groundwater inflow but less on dispersion.

Calibrating the TSM with both tracers increased certainty in model parameters compared to using chloride and radon individually. This is evident in values of the Shannon entropy of the model parameters, which show that the posterior distributions became narrower ($H(X|(radon \cap chloride)) < H(X)$; (Table 2)). Similarly, calibrating the TSM with each tracer individually increased certainty in the model parameters ( $H(X|radon) < H(X)$ and $D_{KL}(X|(radon), X) > 0$ and ($H(X|chloride) < H(X)$ and $D_{KL}(X|(chloride), X) > 0$).





**Table 2: Shannon entropy *H* and Kullback-Leibler divergence $D_{KL}$ for the prior and posterior distributions of model parameters (*D*, $A_{TS}$, $\alpha$ and the groundwater inflow $q_I$) resulting from joint and individual calibration of the TSM with chloride and radon. Results from all three calibration approaches are shown here, which differ in how groundwater inflow was calibrated ($Q_{fix}$, $Q_{LHS}$, and $Q_{out}$). For simplicity, only the results of the top 10% behavioural parameter sets from the low-degassing model setup ($k_{low}$) with radon here for reach #1. Results for reaches #2 - #5 can be found in the supporting information (Table S1).**

| Calibration approach | $Q_{fix}$ | | | | | $Q_{LHS}$ | | | | | $Q_{out}$ | | | | |
|---|---|---|---|---|---|---|---|---|---|---|---|---|---|---|---|
| Parameter | D | α | $A_{TS}$ | $q_I$ | SUM | D | α | $A_{TS}$ | $q_I$ | SUM | D | α | $A_{TS}$ | $q_I$ | SUM |
| Unit | [m² s⁻¹] | [s⁻¹] | [m²] | [m³ s⁻¹ m⁻¹] | | [m² s⁻¹] | [s⁻¹] | [m²] | [m³ s⁻¹ m⁻¹] | | [m² s⁻¹] | [s⁻¹] | [m²] | [m³ s⁻¹ m⁻¹] | |
| $H(X)^a$ | 3.91 | 3.91 | 3.91 | - | | 3.91 | 3.91 | 3.91 | 0 | | 3.91 | 3.91 | 3.91 | 3.90 | |
| $H(X\|chloride)$ | 2.55 | 3.81 | 3.53 | - | | 2.14 | 3.86 | 3.65 | 0 | | 2.16 | 3.86 | 3.70 | 3.87 | |
| $H(X\|radon)$ | 3.88 | 3.74 | 1.65 | - | | 3.89 | 3.88 | 3.39 | 0 | | 3.90 | 3.90 | 3.75 | 2.49 | |
| $H(X\|(radon \cap chloride))$ | 2.55 | 2.40 | 1.51 | - | | 2.00 | 3.72 | 3.72 | 0 | | 1.99 | 3.81 | 3.83 | 1.93 | |
| $D_{KL}(X\|chloride,X)$ | 1.36 | 0.10 | 0.38 | - | 1.84 | 1.77 | 0.06 | 0.27 | 0 | 2.1 | 1.76 | 0.04 | 0.21 | 0.04 | 2.05 |
| $D_{KL}(X\|radon,X)$ | 0.03 | 0.17 | 2.27 | - | 2.47 | 0.01 | 0.03 | 0.52 | 0 | 0.56 | 0.01 | 0.02 | 0.16 | 1.42 | 1.61 |
| $D_{KL}(X\|(radon \cap chloride),X)$ | 1.95 | 1.50 | 2.40 | - | 5.85 | 1.91 | 0.23 | 0.19 | 0 | 2.33 | 1.92 | 0.10 | 0.08 | 1.98 | 4.08 |


### 3.3 Parameter sensitivity and certainty

Parameter sensitivity and certainty depended on the tracer used for calibration. When the TSM was calibrated with chloride, dispersion showed high sensitivity, whereas groundwater inflow was largely insensitive (Table 3). Calibrating the TSM with radon resulted in no sensitivity to dispersion but high sensitivity to groundwater inflow. The sensitivity and certainty of $A_{TS}$

and α in simulations depended both on the tracer used for calibration and the approach used to calibrate groundwater inflow (Fig. 2, Fig. 3, Fig. 4, Table 3, Table S2). Calibration with chloride consistently resulted in high parameter certainty and moderate to high sensitivity for $A_{TS}$ and α. In contrast, calibration with radon yielded moderate to high sensitivity of $A_{TS}$ and α when groundwater inflow was fixed ($Q_{fix}$). $A_{TS}$ and α became insensitive when groundwater inflow was included as a direct calibration parameter ($Q_{LHS}$) or calculated from calibrated discharge ($Q_{out}$). A comparison of model performance showed that

the nRMSE values of behavioral parameter sets were lower when calibrating with radon than with chloride (Fig. 2).





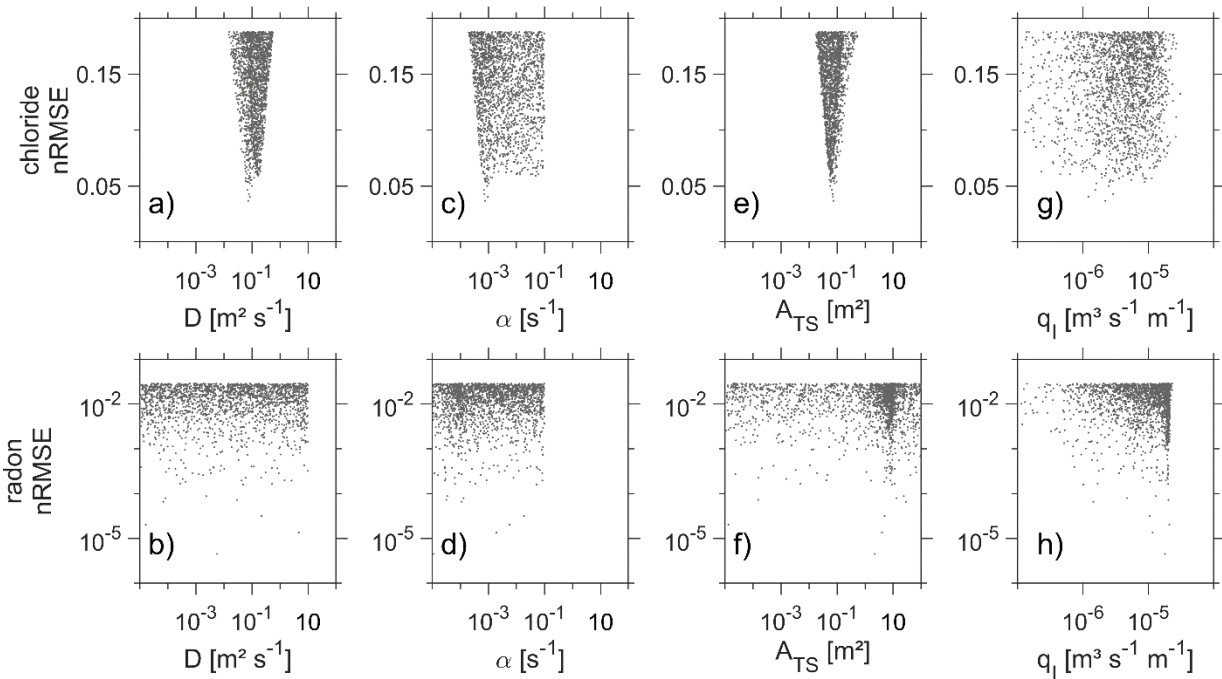

**Figure 2: Parameter values ($D$, $\alpha$, $A_{TS}$, and the groundwater inflow ($q_I$)) plotted against the corresponding normalized root mean square error (nRMSE) values. The figure shows the best 1% of model runs from calibrating the TSM with chloride (top row) and radon (bottom row) using the calibration approach in which groundwater inflow $q_I$ was calculated from calibrated discharge ($Q_{LHS}$).**

**The model runs shown are from reach #1. For radon, we present the $k_{low}$ model setup only.**





**Table 3: Overview of the sensitivity based on the K-S test for all model parameters ($D$, $\alpha$, $A_{TS}$, and the groundwater inflow ($q_I$)) for calibrating the TSM with chloride or radon. Results are presented for all reaches, but only for the 1% behavioral parameters. For simplicity, we show the $k_{low}$ model setup for calibrating the TSM with radon only.**

|  | parameter | $Q_{fix}$ chloride | $Q_{fix}$ radon | $Q_{LHS}$ chloride | $Q_{LHS}$ radon | $Q_{out}$ chloride | $Q_{out}$ radon |
|---|---|---|---|---|---|---|---|
| Reach #1 | $D$ | high | slight | high | slight | high | slight |
|  | $\alpha$ | high | high | high | slight | high | slight |
|  | $A_{TS}$ | high | high | high | high | high | moderate |
|  | $q_I$ |  |  |  |  | slight | high |
| Reach #2 | $D$ | high | slight | high | slight | high | moderate |
|  | $\alpha$ | high | moderate | high | slight | high | insensitive |
|  | $A_{TS}$ | high | high | high | moderate | high | high |
|  | $q_I$ |  |  |  |  | high | high |
| Reach #3 | $D$ | high | insensitive | high | insensitive | high | slight |
|  | $\alpha$ | high | high | high | slight | high | slight |
|  | $A_{TS}$ | high | high | high | high | high | moderate |
|  | $q_I$ |  |  |  |  | moderate | high |
| Reach #4 | $D$ | high | slight | high | slight | high | insensitive |
|  | $\alpha$ | high | high | high | moderate | high | slight |
|  | $A_{TS}$ | high | high | high | high | high | moderate |
|  | $q_I$ |  |  |  |  | slight | high |
| Reach #5 | $D$ | high | insensitive | high | slight | high | moderate |
|  | $\alpha$ | high | high | high | slight | moderate | insensitive |
|  | $A_{TS}$ | high | high | high | moderate | high | high |
|  | $q_I$ |  |  |  |  | high | high |






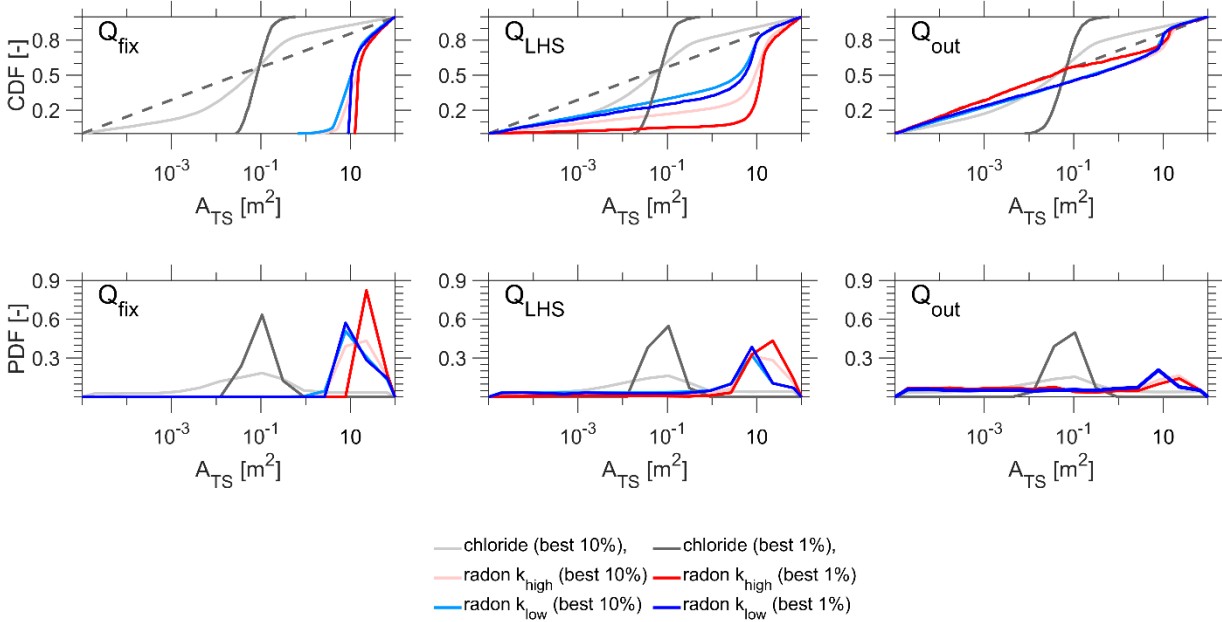

**Figure 3: Cumulative distribution plots (upper row) and posterior distribution plots (lower row) for $A_{TS}$ based on the best 1% and best 10% parameter sets. We present results from the calibration approach where groundwater inflow was fixed ($Q_{fix}$), calculated from calibrated discharge ($Q_{LHS}$), and directly calibrated ($Q_{out}$). Non-behavioral $A_{TS}$ parameters in the top row are shown with a grey dashed line. For simplicity, we show the results from reach #1 only.**






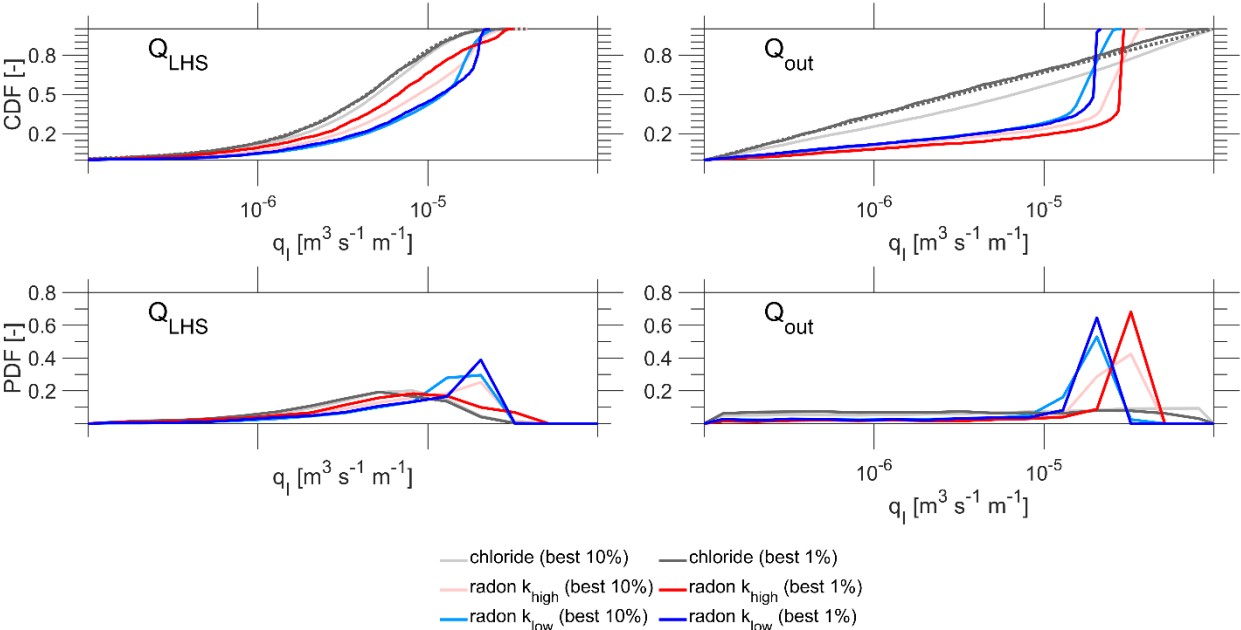

**Figure 4: Cumulative distribution plots (upper row) and posterior distribution plots (lower row) for the groundwater inflow ($q_I$) for the best 1% and best 10% parameter sets. We present results from the calibration approach where groundwater inflow was fixed ($Q_{fix}$), calculated from calibrated discharge ($Q_{LHS}$), and directly calibrated ($Q_{out}$). Non-behavioral $q_I$ parameters in the top row are shown with a grey dashed line. For simplicity, the results from reach #1 are shown only.**

## 3.4 Parameter interactions

We found no parameter interactions when the TSM was calibrated with chloride, with only a few exceptions (Fig. 5, Table S3). Calibration with chloride resulted in a narrower range between the minimum and maximum values of the behavioral parameters, thereby better constraining parameter values. In contrast, when the TSM was calibrated with radon, parameters were tightly constrained only when others in the same set were less constrained (Fig. 5). For example, groundwater inflow was tightly constrained at higher values within the behavioral parameter range, but $A_{TS}$ values for these groundwater inflow values remained unconstrained. Conversely, when $A_{TS}$ values were tightly constrained at higher values, groundwater inflow values associated with them were unconstrained.



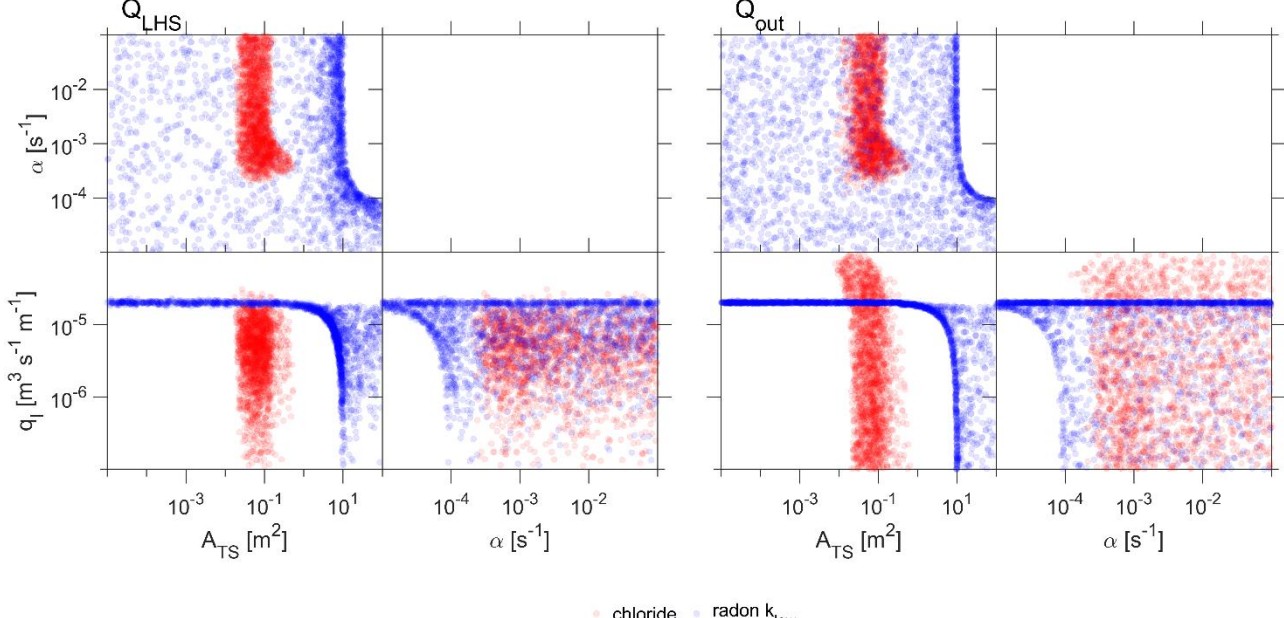

**Figure 5: Scatter plots of the best 1% model parameters ($A_{TS}$, $\alpha$ and $q_I$) from calibrating the TSM with radon (blue) and chloride (red) individually. Calibration with chloride resulted in constrained parameter values. When the TSM was calibrated with radon, parameter values were tightly constrained only when others in the same set were less constrained. We show the calibration approach where the groundwater inflow was calculated from calibrated discharge ($Q_{LHS}$) and the calibration approach where the groundwater inflow was directly calibrated ($Q_{out}$). For simplicity, only the results from calibrating the TSM with radon in the $k_{low}$ model setup are shown. The parameter $D$ is not included, as it was neither certain nor sensitive in calibrating the TSM with radon.**

## 3.5 The effect of different locations of groundwater inflow on parameter interactions

We found significantly different distributions of the behavioral parameters depended on the different locations of groundwater inflow (upstream-most point, mid-point, downstream-most point model setups) independently of which tracer was used for calibration (Fig. 6, Fig. 7). Calibrating the TSM with chloride resulted in constrained parameters when the inflow was located at the upstream-most point or mid-point (Fig. 7). In contrast, parameter interactions became evident when inflow was at the downstream-most point. When the TSM was calibrated with radon and groundwater inflow was set at the upstream-most point, $A_{TS}$ was tightly constrained at higher values within the behavioral parameter range. However, the groundwater inflow values remained unconstrained for these higher $A_{TS}$ values. In contrast, parameter values were less constrained when inflow was set at the downstream-most point.



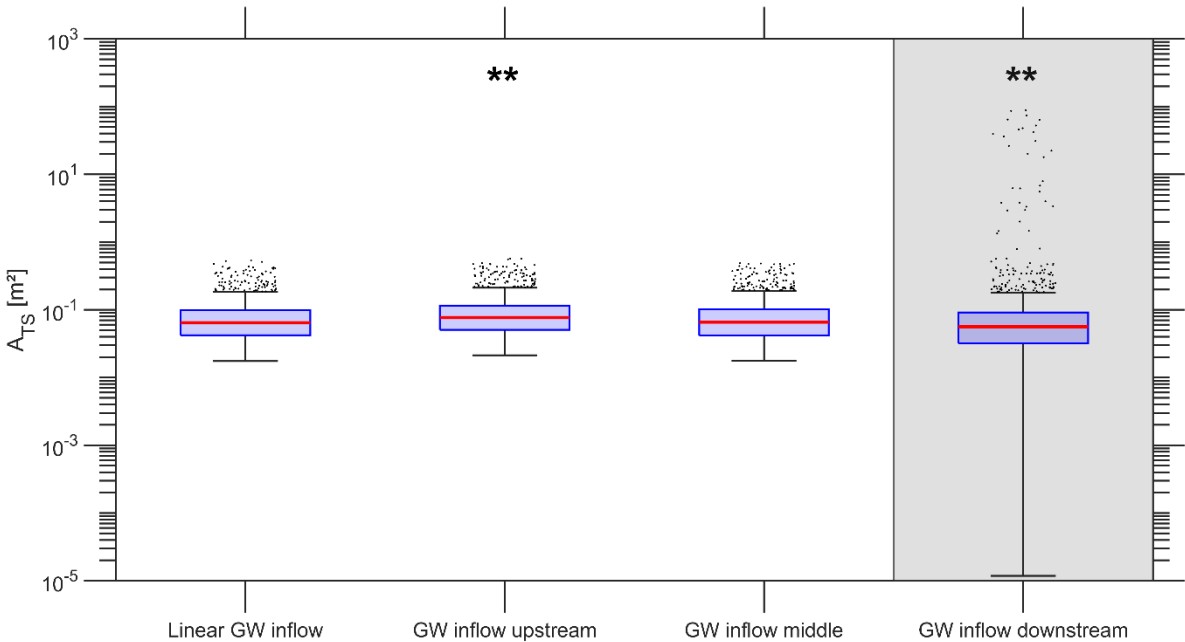

**Figure 6: Distributions of behavioral $A_{TS}$ values from model setups that vary in groundwater inflow location and calibrated with chloride. The model setup labelled 'linear groundwater (GW) inflow' refers to the calibration approach where the groundwater inflow was calculated from the calibrated discharge ($Q_{LHS}$). The red line is the median of the distributions, while black dots highlight outliers. Asterisks show a significant difference between the variance of the parameter distributions compared to the setup with linear groundwater inflow. Results are shown for the best 1% behavioral model parameters and reach #1 only.**





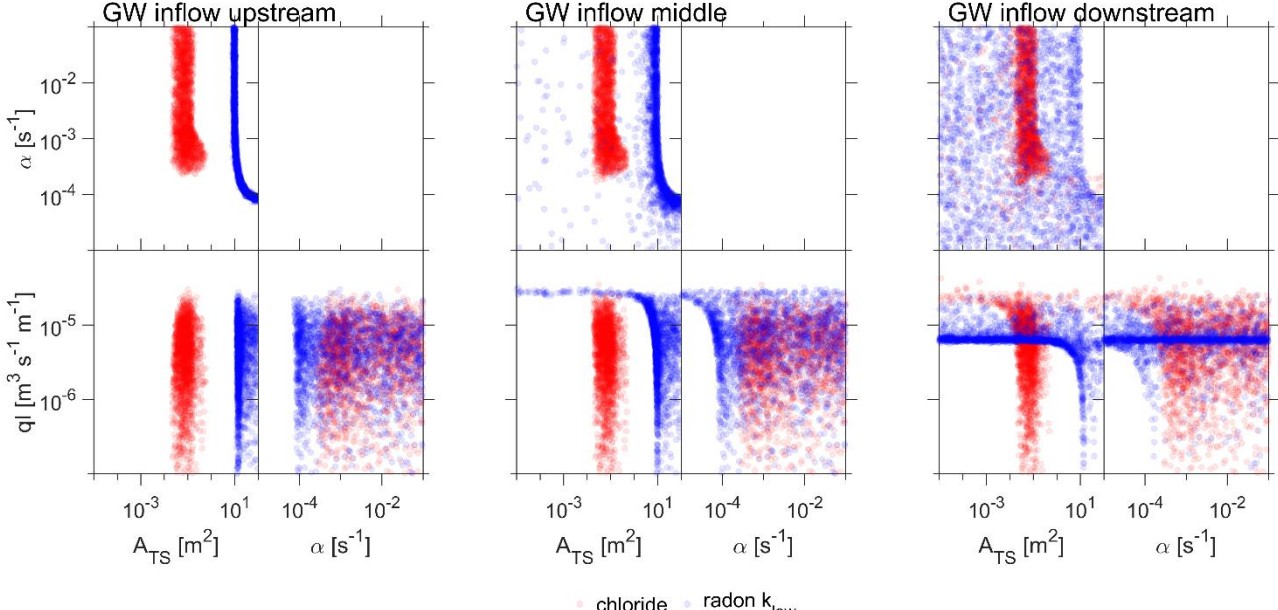


**Figure 7: Scatter plots of the best 1% behavioral model parameters ($A_{TS}$, $\alpha$ and the groundwater inflow $q_I$) from calibrating the TSM with radon and chloride individually. Three different model setups are presented that differ in the location of the groundwater inflow along the reach (location upstream, middle and downstream). For simplicity, results from calibrating the TSM with radon in the $k_{low}$ model setup are shown only.**


## 3.6 Gross water fluxes from calibrating the TSM with radon and chloride

Discharge gradually increased in the downstream direction and ranged from 9.5 to 12 L s$^{-1}$ across all reaches (Fig. 8a). However, not all reaches exhibited this increase; specifically, discharge decreased in reaches #1 and #4. Gross loss and gain revealed spatial variability across reaches. Reaches #1 and #4 were characterized by higher gross losses compared to remaining

reaches (Fig. 8b). Notably, gross water flux could only be derived from calibrating the TSM with radon and not with chloride.



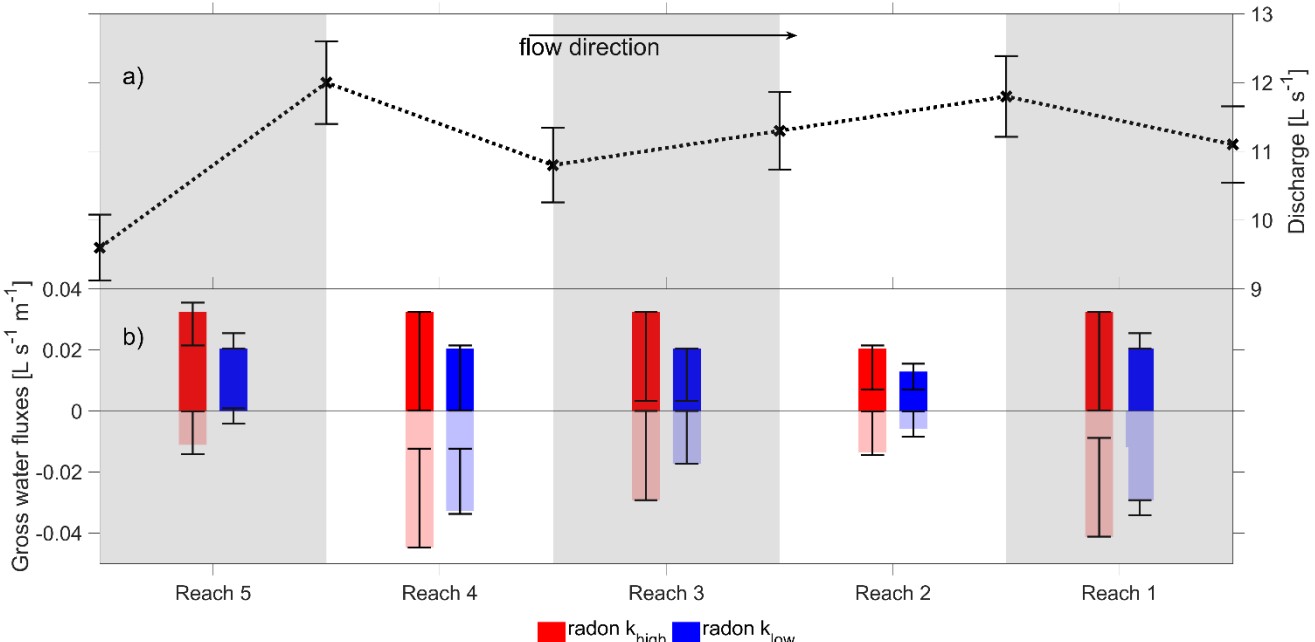

**Figure 8: a) Discharge values upstream and downstream of each reach [L s⁻¹] with 95% confidence intervals. Discharge was calculated from BTCs using dilution gauging. b) Gross water fluxes [L s⁻¹ m⁻¹] for each reach, derived from calibrating the TSM**
**with radon measurements. Both degassing model setups are shown (blue and red) and transparent bars highlight negative gross water fluxes. Bar heights correspond to the mode of the posterior distribution for the calibrated $q_I$ and calculated $q_{out}$ values. Error bars show the 95% confidence interval of the posterior distribution.**

## 4 Discussion

### 4.1 Calibrating TSMs with multiple tracers better constrains model parameters

Calibrating the TSM with chloride provides more information on solute transport (i.e., the sum of the Kullback-Leibler divergence for all TSM parameters) than calibrating it with radon, as chloride provides more information on the dispersion parameter (Table 2). Previous studies have shown that dispersion mainly affects the rising limb of BTCs (e.g., Kelleher et al., 2013; Scott et al., 2003; Wlostowski et al., 2013), suggesting that tracers with a distinct rising concentration limb, such as

chloride, are necessary to identify dispersion. Radon activity remains steady throughout the experiment, which limits its ability to identify dispersion. Although radon yields less information on dispersion than chloride, radon provides more information on groundwater inflow and $A_{TS}$ (Table 2). This is due to the higher radon activity in groundwater compared to surface water, with a 23-fold difference at Oak Creek, which increases radon activity in streams. In contrast, chloride concentrations are smaller in groundwater compared to surface water. Groundwater inflow therefore dilutes chloride concentrations in the stream

without providing additional information on the inflow itself. Ultimately, the unique information each tracer provides about



different model parameters enhances the certainty of solute transport when the TSM is jointly calibrated with radon and chloride compared to with either tracer individually. We therefore recommend calibrating TSMs with multiple tracers to improve estimates of solute transport in future studies. This recommendation aligns with recent calls for joint calibration of hydrological models with multiple tracers. For example, Neilson et al. (2010b) demonstrated that calibrating a TSM with both

temperature and slug tracer data provides more insights into solute transport and exchange compared to calibrating the TSM with temperature individually. At the catchment scale, Rodriguez et al. (2021) demonstrated that jointly calibrating a storage selection function with deuterium and tritium improved certainty on model parameters compared to calibrating the storage selection function with either tracer individually. Notably, the authors employed a different quality criterion for behavioral parameter selection than we did. Rodriguez et al. (2021) used distinct threshold values for each tracer to obtain a comparable

number of behavioral parameter sets, as the sampling frequency - and thus the dataset length - differed between deuterium and tritium. We selected the best 1% and 10% of parameter sets as behavioral to maintain consistency with previous solute transport studies (e.g., Kelleher et al., 2019; Wagener et al., 2002; Ward et al., 2013, 2017; Wlostowski et al., 2013). Selecting the best 1% and 10% of parameter sets led to lower nRMSE for radon compared to chloride (Fig. 2), meaning that calibrating the TSM with radon in our study carried more weight in the joint calibration. Therefore, the conclusion that jointly calibrating the TSM

with radon and chloride yields more information than calibrating it with either tracer individually is influenced by the quality criterion for parameter sets, and thus by subjective modeling decisions - a well-known challenge in hydrology (e.g., Beven and Binley, 1992).

## 4.2 The role of groundwater inflow for parameter identifiability

The sensitivity of radon to the amount and location of groundwater inflow hinders the derivation of narrow, well-constrained estimates for groundwater inflow, $A_{TS}$, and $\alpha$, when calibrating TSMs with radon individually. The sensitivity to the amount of groundwater inflow is evident in the calibration approach where groundwater inflow was calculated from calibrated discharge ($Q_{LHS}$; Fig. 5). Discharge was sampled within the uncertainty range of measurements (95% confidence interval from dilution gauging). Even within this uncertainty range, different values for the groundwater inflow led to unconstrained values

of $A_{TS}$ and $\alpha$. This suggests that obtaining narrow, well-constrained estimates for groundwater inflow, $A_{TS}$, or $\alpha$ from calibrating the TSM with radon will remain challenging unless at least one of these parameters is further constrained. Deriving narrow, well-constrained TSM parameters is also restricted due to the sensitivity of radon to the location of groundwater inflow. This is evident in different parameter interactions across model setups that vary in the location of groundwater inflow (Fig. 7). For example, groundwater inflow at the downstream-most point in the study reach increases radon activity at this point, where

measured radon values are used for calibration. A shorter distance between the inflow and the downstream-most point allows less time for radon to degas. With less time for degassing, only smaller, better constrained values of groundwater inflow can close the radon mass balance and result in a good fit between simulated and measured radon activity. Constrained groundwater inflow values lead to less-constrained estimates of $A_{TS}$ and $\alpha$, as different values of $A_{TS}$ and $\alpha$ can still close the radon mass



balance. Thus, the spatial variability of groundwater inflows hinders better constraints on parameter estimates for $A_{TS}$ and $\alpha$
when using radon individually for model calibration.

Spatially heterogeneous groundwater inflows have been documented across various streams and attributed to transitions in valley structure (Mallard et al., 2014; Cartwright and Gilfedder, 2015; Payn et al., 2009; Pittroff et al., 2017; Somers et al., 2016), geological fractures (Genereux et al., 1993; Glaser et al., 2020), or textural heterogeneities of the subsurface textural heterogeneities (Fleckenstein et al., 2006). Groundwater inflows at discrete locations of streams are common (Sophocleous
2002), yet their impact on the identifiability of TSM parameters with radon and chloride has not been explored. Instead, previous research has highlighted the critical role of degassing when simulating radon activity (e.g., Atkinson et al., 2015; Gilfedder et al., 2019). For example, Schubert et al. (2020) incorporated degassing tests alongside radon measurements to quantify groundwater inflow through a numerical mass balance approach with transient storage parameters (Frei and Gilfedder, 2015). Schubert et al. (2020) found that calibrated groundwater inflows exceeded the net increase in discharge, attributing this
outcome to spatial variability in degassing rates. The two degassing parameterizations in our study did not affect model performance (e.g., Fig. 3 and 4); rather, differences in groundwater inflow locations along the reach resulted in different model parameters in behavioral parameter sets. Despite differences in our study site and that of Schubert et al.'s (2020), both models assume linear groundwater inflow along the reach. Therefore, the observed overestimation of calibrated groundwater inflows by Schubert et al. (2020) may also result from spatial variability in groundwater inflows along their study reach, leading to
radon increases that required calibrating higher groundwater inflow values. Similarly, Cook et al. (2006) showed an overestimation of groundwater inflow to the Cockburn River, Australia, by almost 70% compared to actual flow measurements. The authors concluded that including exchange with subsurface transient storage zones is essential when simulating radon activity. Cook et al. (2006) assumed linear inflow along the reach, similar to Schubert et al. (2020). In light of our findings, the overestimation found by Cook et al. (2006) may also be due to spatial variability in groundwater inflows
rather than the omission of exchange with subsurface transient storage zones in their radon mass balance. The insights from our study are therefore critical for contextualizing results in future radon-based studies in river corridors, especially given the prevailing perception in the radon community – based on Cook et al.'s (2006) findings nearly twenty years ago – that exchange with subsurface transient storage zones should be included in radon mass balances to close them.

Notably, our results show that spatially heterogeneous groundwater inflows also affect the identifiability of $\alpha$ and $A_{TS}$ when
calibrating the TSM with chloride concentrations (Fig. 6). Previous studies have noted uncertainty of the $\alpha$ estimates from the behavioral parameters when calibrating TSMs with chloride (Kelleher et al., 2013; Wagener et al., 2002; Wlostowski et al., 2013). These studies attributed the uncertainty of $\alpha$ to differences in stream-specific characteristics. Our findings offer a more detailed explanation, namely the spatial variability in groundwater inflow along streams. We therefore suggest that further research is needed to understand how groundwater inflow influences chloride concentrations and, in turn, the identifiability of
parameters when calibrating the TSM. Future studies on solute transport with chloride should first identify groundwater inflow



locations before selecting a reach for slug tracer injections. This could be achieved by incorporating spatially resolved piezometers and hydraulic head measurements along streams, as implemented in the study design of Harvey (1993) and Bonanno et al. (2023). Alternatively, temperature surveys (distributed temperature sensing; 'DTS'), which provide high spatial and temporal resolution data on longitudinal groundwater inflows (Krause et al., 2012), could be used.


### 4.3 Radon is biased towards large spatial-scale subsurface flow paths

Spatially variable gross water gains exceeding net discharge from calibrating the TSM with radon suggest that Oak Creek's water balance is influenced by stream–subsurface exchanges occurring at multiple spatial scales. Previous studies have shown that large spatial-scale subsurface flow paths play a critical role in explaining water mass balances in streams (e.g., Payn et al.,

2009; Stanford and Ward, 1993; Ward et al., 2023). The inability to detect these gross water gains with chloride due to nearly complete BTC mass recovery highlights radon's superior sensitivity in revealing large-scale subsurface flow paths. This is because radon activity was in steady state at both the upstream and downstream end of the reach before the slug tracer experiment began. Steady-state radon activity indicates that the ensemble of timescales for many parcels of water in the subsurface is labelled with radon prior to the experiment. Subsurface flow paths originating from further upstream also

contribute to radon activity at the downstream end of the reach (Fig. 9, arrow D), alongside contributions from groundwater inflow and temporally shorter flow paths (Fig. 9, arrows A, B, and E; Cook et al., 2013; McCallum et al., 2012). In contrast, chloride is injected as a distinct input signal (Dirac injection) at the upstream end of the reach, thus only revealing flow paths between locations of injection and measurement (Fig. 9, arrows A and B). This finding underscores radon's value as a tracer for detecting large spatial-scale subsurface flow paths that slug injections of chloride may overlook. However, large spatial-

scale subsurface flow paths are not explicitly represented in the TSM. Instead, the flow paths are indirectly accounted for through the calibration of the groundwater inflow. This indirect consideration may lead to an overestimation of groundwater inflow values, which reduces the certainty of the $A_{TS}$ and $\alpha$ parameters and introduces interactions between the groundwater inflow, $A_{TS}$, and $\alpha$. Although radon's sensitivity to large spatial-scale subsurface flow paths provides valuable insights, it introduces bias in constraining transient storage parameters in TSMs. Thus, radon traces subsurface timescales longer than the

WoD, which cannot be identified through calibrating the TSM with radon.





**Figure 9: Conceptual figure of flow paths in streams. The box shows the reach of investigation. A is a flow path that is labelled by**
**the tracer at the upstream location and returns within the WoD. B is a flow path with timescales longer than the WoD, where transit**
**times exceed the duration of the tracer experiment, as indicated by the dashed arrow. C is a flow path that bypasses the downstream**
**end of the reach. D is a subsurface flow path entering the hyporheic zone upstream of the reach. This path is characterized by transit**
**times shorter than 21 days and radon activity that has not yet reached secular equilibrium. E is a groundwater flow path with transit**
**times longer than 21 days and radon activity at secular equilibrium. Flow path D is conceptually excluded from the TSM, which is**
**why it is highlighted with a red dashed arrow.**

For future research with radon on gross exchange fluxes, we recommend constraining the transient storage parameter by

considering surface (e.g., pools) and subsurface transient storage zones (e.g., hyporheic zones) in TSM evaluations (e.g., Choi

et al., 2000). This step might be critical, given that only surface storage contributes to radon degassing. OTIS-R currently

considers a single storage zone for the $A_{TS}$ parameter, which actually includes both surface storage and subsurface storage.

Moreover, we recommend that future studies incorporate additional discharge measurements to reduce measurement errors.

This is necessary because the calibrated gross exchange fluxes for Oak Creek fall within the confidence interval of changes in



net discharge (Fig. 8). Therefore, caution is needed when interpreting these fluxes. By addressing these recommendations, future studies may unlock radon's full potential as a tracer, enabling deeper insights into the continuum of subsurface flow paths beyond the WoD of slug tracer injections. Experimental setups could for instance measure radon activity alongside BTCs at several downstream locations from the point of injection (e.g., Ward et al., 2023), thereby supporting the assessment of large spatial-scale subsurface flow paths from slug tracer injections. Studies focusing on understanding the turnover of chemical compounds along large spatial-scale subsurface flow paths could benefit from incorporating radon as an additional tracer. This is especially relevant for slower chemical degradation processes, such as denitrification, which are more likely to occur along flow paths with longer timescales (sensu Jimenez-Fernandez et al., 2022) which radon may help resolve.

## 5 Conclusion

The overarching goal of this study was to quantify flow paths of different timescales at the reach scale using measurements of solute tracer and naturally occurring radon. To achieve this, we calibrated a transient storage model (TSM) with both tracers individually and jointly. Our results show that calibrating the TSM with chloride yields identifiable, narrower parameter values, while calibration with radon do not result in identifiable parameters. The lack of identifiability with radon is due to steady-state activity of radon in the stream, the tracer's sensitivity to the amount and location of groundwater inflow, and to large-scale subsurface flow paths, demonstrating radon's bias toward flow paths with longer temporal timescales. We found that joint calibration of a TSM with both tracers provided the most information on parameters of the TSM compared to calibrating it with either tracer individually. Based on these findings, we recommend that future studies incorporate both radon and chloride when calibrating the TSM to improve estimates of solute transport. Furthermore, with an adapted sampling and modeling strategy, we see significant potential in incorporating radon in future studies to unlock its full potential in revealing flow paths with longer timescales than those traced by slug tracer injections.

**Code availability**

The code to obtain the identifiability of the transient storage parameter and OTIS-R are freely available at Bacher et al. (2025).

**Data availability**

Solute breakthrough curve and radon activity can be obtained at Bacher et al. (2025).



## 6 Author contribution

All authors developed the concept of the manuscript and discussed the content of this manuscript. Mortimer Bacher took care of the formal analysis, programming the software and writing the first draft. Julian Klaus contributed to the original idea and revised and edited the manuscript. Adam Ward and Catalina Segura revised and edited the manuscript. Jasmine Krause conducted experiments and revised and edited the manuscript. Clarissa Glaser had the original idea for this manuscript, conducted the experiments, took care of writing the first draft, implemented the revisions and administration of the project.

## Competing interests


The authors declare that they have no conflict of interest.

## Acknowledgments

The authors would like to thank Jaime Ortega, Madelyn Maffia, Keira Johnson and Stephen Arthur Fitzgerald for supporting field work and Stephen Good for providing lab space for conducting our lab analysis.

## Financial support


This research was supported by the Argelander Starter Kit of the University of Bonn, the Klaus Tschira Stifung, and the Open Access Publication Fund of the University of Bonn.

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
