# Peer review of "Different tracer, different bias: using radon to reveal flow paths beyond the Window of Detection"

_EGUsphere, 2025_

## Author Comment (AC1)

Please find below our responses to the comments of reviewer 1. Our responses are given in blue.

1. The authors have calibrated one-zone stream transient storage models (TSMs) using two different tracers: salt (NaCl) and Radon. The authors state that pairing salt tracers with Radon increases the window of detection (WoD) by up to 21 days, which is much longer than that of salt. The authors argue that including Radon improves the ability to identify the true mean value of transient storage parameters. This paper will make a nice contribution to the literature by providing new findings into measurement and modeling of stream tracer breakthrough curves.
The presentation is good but there were a few places where this reviewed got confused. To benefit the reader through an improved presentation, below are a few general comments and several specific comments for the authors to consider.

The authors state that jointly calibrating to two tracers is an improvement, which is exciting and novel; however, it is not clear to this reviewer where the joint calibration results are presented. In the figures and captions, it appears results are based on induvial tracers. It would help the reader to specifically state which results are the joint calibration and which are individual.

- *Thank you for these important points. We agree that we have not explicitly mentioned what we define as "joint calibration". We will clarify this aspect in the manuscript. The revised version of the text will read as:*

  *"Since we tested the same combinations of parameter values in the TSM for both tracers, the intersection of the behavioral parameter sets from both tracers reflects the parameter sets obtained when the model is calibrated with both tracers together. This indicates that when the behavioral parameter sets for both tracers are identical, the choice of tracer does not affect estimates of solute transport. We refer to this intersecting set as the 'joint calibration'. In contrast, other parameter sets were included in the behavioral set for only one tracer but not the other. These sets represent parameterizations that result in acceptable model performance for a single tracer, but are not robust in simulating both tracers simultaneously. We refer to these as the 'individual calibration'. The behavioral parameter sets were used for all subsequent analysis and calculations."*

- *Furthermore, we will state clearly in the figure and table captions where we refer to the joint and the individual calibration. For example, in Figure 3 in Table 2, we will specify which of the presented values of the Shannon entropy H and Kullback-Leibler divergence correspond to the joint or individual calibration. The revised captions will read as follows:*

  *"Figure 3: Cumulative distribution plots (upper row) and posterior distribution plots (lower row). Each plot shows $A_{TS}$ results using the best 1% and best 10% parameter sets from the individual calibration. (…)"*

  *"Figure 4: Cumulative distribution plots (upper row) and posterior distribution plots (lower row). Each plot shows groundwater inflow ($q_I$) results based on the best 1% and best 10% parameter sets from the individual calibration. (…)"*

  *"Figure 6: Distributions of behavioral $A_{TS}$ values from model setups with varying groundwater inflow locations, each calibrated with chloride only. (…)"*

  *"Table 2: Shannon entropy H and Kullback-Leibler divergence $D_{KL}$ for the prior and posterior distributions of model parameters (D, $A_{TS}$, α and the groundwater inflow $q_I$)*

*resulting from joint ($H(X|radon \cap chloride)$; $D_{KL}(X|(radon \cap chloride),X)$) and individual ($H(X|chloride)$; $H(X|radon)$; $D_{KL}(X|chloride)$ calibration; $D_{KL}(X|radon)$) of the TSM with chloride and radon. (…).”*

2. If Radon is relatively steady and therefore does not capture the entire BTC, why would one assume that a TSM could be calibrated using only Radon? Pairing salt with Radon creates a more representative BTC. In simple terms, better calibration data, a better chance to identify parameters. That type of statement would benefit the reader.

- *We agree that clearer explanation will be needed regarding the assumption of using radon alone to calibrate the TSM. Conceptually, the underlying assumption is that exchange processes occur regardless of whether tracer input is steady (e.g., from natural tracers like radon) or from a defined injection (e.g., chloride slug or constant-rate injections). From a physical point of view, these processes should label the same flow paths within a reach, leading to similar model parameters independent of the tracer used. This principle has previously been supported by studies simulating transit time distributions at the catchment scale (Rodriguez et al. 2021). We will rewrite part of the introduction to clarify this aspect:*

  *“To answer these questions, we apply a coherent mathematical framework to radon and slug injections of sodium chloride (NaCl). This approach is motivated by previous catchment-scale studies showing that applying the same model framework to different tracers yields comparable insights into hydrological transport processes (e.g., Rodriguez et al., 2021; Wang et al., 2023). To ensure a coherent mathematical framework for both NaCl and radon, we adapt the transient storage model OTIS ('One-Dimensional Transport with Inflow and Storage model,' Runkel, 1998) by incorporating radon-specific processes such as degassing. We then jointly and individually calibrate the model with slug tracer (chloride) and radon data. We perform a global sensitivity analysis approach to assess parameter identifiability. Finally, we apply information theory to quantify the information gained from joint and individual calibration of the TSM with these tracers.”*

- *Thank you also for pointing out the need for more accessible language throughout the manuscript. We will revise the text accordingly to improve clarity and readability.*

3. Parameter "certainty" is difficult for this reviewer to understand and follow because "sensitivity" and "identifiability" seem to already cover that concept. How is certainty different than identifiability? If clearly a different metric, that makes sense. If certainty means something like identifiability, introducing a new term "certainty" just adds confusion.

- *Thank you for pointing this out. We agree that the distinction between identifiability, sensitivity, and certainty requires clarification. In our study, we are specifically interested in parameter certainty, which we define as the ability to estimate a parameter with low uncertainty. A key reason for parameter uncertainty is insensitivity, meaning that the model outcome does not change substantially with the parameter value. In such cases, a wide range of parameter values can produce similarly low nRMSE values. To address this, we first analyze sensitivity using nRMSE-versus-parameter plots and cumulative distribution function (CDF) plots. However, insensitivity is not the only cause of parameter uncertainty. Parameter interactions can also reduce identifiability, which we examine using scatter plots. These analyses allow us to characterize the main drivers of potential uncertainty. Finally, to quantify certainty itself, we analyze posterior distributions and compute Shannon entropy. A parameter is identifiable when it is sensitive, certain, and does not show strong interactions with other parameters. We will clarify this aspect in the text:*

  *"The parameter identifiability analyses include the visual inspection of I) nRMSE vs. parameter plots (Wagener et al., 2003), II) cumulative parameter distribution plots (Kelleher et al., 2019), III) posterior distribution plots (Wagener et al., 2002; Ward et al., 2017), IV) scatter plots of the behavioral parameters, and V) calculation of the Shannon entropy for the posterior distribution of model parameters as metric for certainty (sensu Rodriguez et al. 2021, section 2.6)."*

  *"We used the information content to evaluate parameter certainty by calculating the Shannon entropy of their posterior distributions (Cover and Thomas, 2005; Loritz et al., 2018). Rodriguez et al. (2021) applied this approach in a catchment-scale study. The posterior distribution is the probability density function of the behavioral parameters sets."*

4. If the background was subtracted for salt, why is groundwater influx also needed for the salt TSM? If $Q_{LHS}$ is estimated from discharge, it would help the reader that conditions are still steady state. So groundwater inflow was assumed fixed for all reaches? And then estimated by gage data (dQ/dx). And also estimated directly via calibration. Correct?

- *Subtracting background concentration is common in slug tracer tests, and justified by the peak concentrations significantly exceeding background concentration in the stream. However, inflowing water with less chloride concentration than those measured in the BTC may still have a diluting effect on tracer concentration. Therefore, groundwater influx is also needed for calibrating a TSM with chloride. We will highlight this aspect in the manuscript:*

  *"Although we expect groundwater inflow to primarily affect radon activity in streams, we also calibrated TSM parameters in three model setups that varied in groundwater locations using chloride concentrations. This was motivated by the assumption that chloride-free groundwater, as commonly assumed in TSMs, dilutes chloride concentrations in streams."*

- *Discharge was calibrated in the $Q_{LHS}$ approach rather than kept fixed. We will revise the section describing the calibration procedure to highlight this point. The revised version of the section will read as:*

  *"In the second calibration approach ($Q_{LHS}$), discharge was calibrated as model parameter, like D, α, and $A_{TS}$, before calculating groundwater inflow. For each reach,*

*discharge was sampled from a normal distribution, because Schmadel et al. (2010) reported that discharge measurement errors follow a normal distribution. (…)"*

5. What new information is presented in Figure 9? The red flow path does not seem technically correct.

   – *The novelty of Figure 9 is indeed the flow path "C," which is why it appears in red.*

   should state that this figure has been modified from Payne et al. 2009.

   – *We will clarify that this figure was modified from Payn et al. (2009).*

   Flow path C does not seem technically correct. You do not know if that represents tracer that bypasses measurement site. The WoD is your measurement window, not the real window in which flow paths exist. And why is C red? That adds confusion. This reviewer does not find any new information added by this figure, and it does not seem technically correct. Suggest to revise to clarify which arrow is specifically improved in this study.

   – *Flow path "C" is correctly labeled; it is identified only by radon, not chloride. We will explain the improvements in measuring this flow path and the novel insights provided by the figure throughout the text and in the figure caption:*

   *"Figure 9: Conceptual figure of flow paths in streams. The box shows the reach of investigation. A is a flow path that is labelled by the tracer at the upstream location and returns within the WoD. B is a flow path with timescales longer than the WoD, where transit times exceed the duration of the tracer experiment, as indicated by the dashed arrow. C is a subsurface flow path entering the hyporheic zone upstream of the reach. This flow path is characterized by subsurface transit times shorter than 21 days and radon activity that has not yet reached secular equilibrium. This flow path can be identified by radon but not by chloride concentration, which explains the red coloration. D is a flow path that bypasses the downstream end of the reach. E is a groundwater flow path with transit times longer than 21 days and radon activity at secular equilibrium. Flow path D is conceptually excluded from the TSM, which is why it is highlighted with a red dashed arrow. (Figure adapted from Payn et al. (2009))"*

   *"Spatially variable gross water gains exceeding net discharge from TSM calibration with radon suggest that the water balance of Oak Creek is affected by stream–subsurface water exchange that is occurring at multiple spatial scales. Previous studies have shown that large spatial-scale subsurface flow paths, that are subsurface flow paths originating from further upstream of the stream reach, play a critical role in explaining water mass balances in streams (e.g., Payn et al., 2009; Stanford and Ward, 1993; Ward et al., 2023). Unlike chloride, radon uniquely labels these flow paths (Fig. 9, arrow C), which are otherwise undetectable. This unique labeling underpins the superior sensitivity of radon in revealing subsurface flow paths and highlights its novel contribution to understanding their role in stream water balances. Conceptually, this labeling capacity of radon is reflected in measured steady state activity at both the upstream and downstream ends of the reach before the slug tracer experiment began. Steady state activity indicates that the measured radon activity includes subsurface water parcels pre-labelled with radon prior to the experiment. Thus, large spatial-scale subsurface flow paths from further upstream also contribute to measured radon activity at the downstream end of the reach (Fig. 9, arrow C), along with groundwater inflow and temporally shorter flow paths (Fig. 9, arrows A, B, and E; Cook et al., 2013; McCallum et al., 2012). In contrast, chloride is injected as a distinct input signal (Dirac injection) at the upstream end of the reach and thus only reveals flow paths between locations of injection and measurement (Fig. 9, arrows A and B). This inherent bias in chloride injections toward labeling*

*timescales within the WoD complements the bias of radon toward large spatial-scale subsurface flow paths, underscoring the value of using both tracers together. However, despite these novel insights on large spatial-scale flow paths from calibrating TSM with radon, such paths are not explicitly represented in the TSM. Instead, they are indirectly accounted for through the calibration of groundwater inflow. This indirect consideration may lead to an overestimation of groundwater inflow values during calibration. Such overestimation reduces certainty in the $A_{TS}$ and α parameters and causes interactions among groundwater inflow, $A_{TS}$, and α. Although radon's sensitivity to large spatial-scale subsurface flow paths provides valuable insights, it therefore also introduces bias in constraining transient storage parameters in TSMs."*

6. This reviewer got a bit lost in the Discussion. The alternating representation and descriptions of groundwater and groundwater locations caused a tough read, and the key point seemed buried as a result. Radon helps constrain the GW inflow. Great. The salt is needed for advection and dispersion. Try to write in more simple terms where possible.

   − *Thank you for pointing this out. We will carefully revise the discussion, remove unnecessary repetition of results, and highlight our key findings more clearly. For example, the first section of the discussion will read as:*

   *"Calibrating the TSM with chloride provides more information on solute transport than calibrating the model with radon because chloride is particularly informative on D (Table 2). Previous studies have shown that D mainly affects the rising limb of BTCs (e.g., Kelleher et al., 2013; Scott et al., 2003; Wlostowski et al., 2013). This highlights that tracers with a distinct rising concentration limb, such as chloride, are necessary to identify D. Radon, by contrast, provides more information on groundwater inflow and $A_{TS}$ than chloride (Table 2) due to its higher activity in groundwater compared to surface water. Because chloride concentrations are smaller in groundwater compared to surface water, groundwater inflow simply dilutes chloride concentrations in the stream without adding additional information on inflow. In summary, the most information on solute transport is obtained by using both tracers jointly because each tracer contributions uniquely. Joint calibration with radon and chloride improves certainty in solute transport estimates compared to calibrating with either tracer alone. We therefore recommend calibrating TSMs with multiple tracers to improve estimates of solute transport in future studies.*
   *This recommendation aligns with recent calls for joint calibration of hydrological models with multiple tracers. For example, Neilson et al. (2010b) demonstrated that calibrating a TSM with both temperature and slug tracer data provides more insights into solute transport and exchange compared to calibrating the TSM with temperature alone. At the catchment scale, Rodriguez et al. (2021) demonstrated that jointly calibrating a storage selection function with deuterium and tritium reduced uncertainty in model parameters compared to calibrating the model with either tracer alone. Notably, the authors used a different quality criterion for behavioral parameter selection than we did. Rodriguez et al. (2021) used distinct threshold values for each tracer to obtain a comparable number of behavioral parameter sets due to differences in sampling frequency and thus the dataset length between deuterium and tritium. In contrast, we selected the best 1% and 10% of parameter sets as behavioral to remain consistent with previous solute transport studies (e.g., Kelleher et al., 2019; Wagener et al., 2002; Ward et al., 2013, 2017; Wlostowski et al., 2013). This selection resulted in lower nRMSE for radon compared to chloride (Fig. 2). Thus, TSM calibration with radon carried more weight in the joint calibration. Therefore, the conclusion that joint calibration of the TSM with radon and chloride yields more information than calibration with either tracer alone is partly influenced by the quality criterion for*

*parameter sets and, ultimately, by subjective modeling decisions - a well-known challenge in hydrology (e.g., Beven and Binley, 1992)."*

7. This reviewer feels that some sort of concise recommendation from the authors would be helpful for the reader. How would this transfer to other streams or watersheds? Low versus high background Radon? This reviewer assumes this tracer approach is limited by river size; perhaps it only really applies to smaller headwater type streams? What are the limitations if streamflow is dynamic? Would this study need to be repeated for other streamflow conditions? A reader would benefit from a concise statement.

  – *Thank you for this important point. We will revise the last part of our discussion section and add a subheader titled "Implications." In this new section, we will explicitly explain how to apply these findings to other streams. We will also provide detailed recommendations for future research. Please see our answer to comment no. 33.*

  Citations look mostly complete and are well thought out. Equations appear correct. Nice work overall.

  - *Thank you!*

Specific comments:

8. 14: How do you know a longer estimated timescale would lead to a larger volume? Longer timescale does not necessarily mean more volume.

   – *We will delete this aspect.*

9. 35: consider Schmadel et al. https://doi.org/10.1002/hyp.9854 for supporting definition of WoD.

   - *Thank you for this suggestion.*

10. 58: "slug", suggest to define as a near instantaneous injection of mass.

    - *Thank you for this suggestion. We will define the term "slug":*

      *"This limitation is critical, as a growing body of research highlights the presence of flow paths that exceed the duration of experiments using instantaneous tracer injections (hereafter 'slug tracer experiments'; e.g., Ward et al., 2023)."*

11. 62: Do not follow what "they" are. Exchange fluxes?

    - *We will replace the term "they" by "these subsurface flow paths" to clarify this aspect.*

12. 67: Sentence needs another look. Redundant information, consider delete.

    - *There is a small grammatical error in the sentence, which we will correct. However, we think this sentence is important here, as it provides a logical bridge between the preceding and following paragraphs.*

13. 69: switching to "duration" adds confusion for reader. You specifically mean the WoD, correct? Suggest to state the WoD for consistence as that is clearly defined. Duration of any slug is infinity; the WoD defines which portion of the slud returns real information.

    - *We agree and will change the term "duration of slug tracer experiments" to WoD.*

14. 84: The goal is…there are several places where qualifying words like "overarching" add to the word count and not needed.

    - *Thank you for this important comment. We will carefully revise the entire manuscript and removed qualifying words from the text.*

15. 150" "derived"? Confusing. Solute transport is simply parameterized. You are not necessarily deriving anything new.

    - *Thank you, we will revise the sentence and write:*

      *"Solute transport parameters are commonly determined by calibrating TSMs against measured BTCs."*

16. 153: well or a "completely mixed" transient storage zone.

    - *We agree and will revise as suggested.*

17. 162: fine, but mention of the ADE if no storage is not necessary for the reader.

  - *Thank you for this suggestion. We will delete the sentence "In case of a lack of exchange where α = 0, eq. 1 reduces to the advection-dispersion equation."*

18. 163: the model formulation was further modified to account for gas exchange. "Not suited" reads awkwardly.

  - *We will remove the term "not suited for" and change the sentence to "The model formulation above does not account for key processes affecting radon activity,…"*

19. 253: "certainty" adds confusion for this reviewer. H is entropy, but you are also calling that "certainty." This reviewer has referred to parameter sensitivity and identifiability, but not certainty.

  – *We will revise the definition of certainty in the text (see response to comment no. 3).*

20. 300: Runkel and Chapra 1993 should probably be cited here. Also, important to point out for the reader that this is steady state, meaning flow is considered steady for every slug injection. If dynamic, dA/dt + dQ/dx = qi = qout. Also, it is not clear what was needed for the TSM. You only need this equation to estimate qi and plug into the TSM. And what was the Radon concentration assumed in GW? Does qi not matter for salt because the background was subtracted?

  – *Thank you for raising this concern. We will state that this applies under steady-state conditions. The revised sentence will read as follows:*

    *"This is because OTIS, and by extension OTIS-R, accounts for water mass balance under steady state by parameterizing groundwater inflow using the following equation: (…)"*

  – *Furthermore, we will report the radon activity used for lateral groundwater inflow:*

    *"For all three calibration approaches, the measured equilibrium radon activity was used as the activity of the groundwater inflow."*

21. 320: Reads as flat. A reader is left wondering what the key result is. Perhaps say that tracer results were ideal for testing this approach because there was a clear difference between surface water and groundwater concentrations. "tracers revealed spatial variability" is not necessarily novel. Suggest to add a concise statement to help the reader regarding what you see in the tracer data. For example, "pairing two tracers allowed for improved identification of parameters because surface water and groundwater are distant in this watershed." Something like that.

  - *Thank you for pointing this out. We will rewrite the section as follows:*

    *"BTCs (Fig. S1) showed a distinct peak concentration at both the upstream and the downstream locations of the study reaches, thereby meeting a key requirement for the calibration of the TSM. Radon activity in groundwater was 23 times higher than in surface water, providing the necessary contrast for quantifying groundwater inflows into the stream. Stream radon activity ranged from 285 (± 22) Bq m$^{-3}$ to 337 (± 26) Bq m$^{-3}$ with the highest activity was observed at reach #2."*

22. 334: Increased certainty? You mean improved parameter identifiability? It is not clear how certainty and identifiability are different or same. Same comment for line 348.

   - *We will explain in detail what we mean by certainty (see our response to the comment no. 3)*

23. 360: In the caption, it would help the reader to specify these are two separated calibrations with an "only salt" and "only radon"

   - *We will revise as suggested.*

24. 365: unclear what Qfix, QLHS, Qout are directly from the table. So ATS is highly sensitive to Radon tracer?

   - *We will clarify these aspects by revising the table caption. The caption will define the terms 'high', 'moderate', 'poor', and 'non-sensitive' for clarity. For consistency, we will also align the terminology used in the table with that introduced in the Methods section. The revised caption will read as follows:*

     *"Table 3: Overview of the sensitivity based on the K-S test for all model parameters (D, α, $A_{TS}$, and the groundwater inflow ($q_I$)) from the individual calibration of the TSM with chloride and radon. The terms 'high', 'moderate', 'poor', and 'non-sensitive' refer to the classification of parameter sensitivity based on the K-value and p-value (see Section 2.6), following the methodology of Ouyang et al. (2014). Results from all three calibration approaches are shown here, which differ in how groundwater inflow was calibrated ($Q_{fix}$, $Q_{LHS}$, and $Q_{out}$). Results are presented for all reaches, but only for the 1% behavioral parameters. For simplicity, we show the klow model setup for calibrating the TSM with radon only."*

25. 410: This reviewer does not follow the main point or reason for this Figure 6, and finds this figure confusing. The GW categories are not the same as previous discuss in text. Qfix, QLHS, Qout. What is GW inflow downstream? And why is that shaded grey. The point is a few outliers are caused? Medians and percentiles are nearly the same.

   – *Thank you for this important point. We will use the same terminology as in the previous text for consistency. We will also clarify the purpose of the shaded grey area.*

[Figure]

*"Figure 6: Distributions of behavioral ATS values from model setups with varying groundwater inflow locations, each calibrated with chloride only. The model setup labelled 'linear groundwater (GW) inflow' refers to the calibration approach where the groundwater inflow was calculated from the calibrated discharge (QLHS). The red line is the median of the distributions, while black dots highlight outliers. Asterisks and the grey areas show a significant difference between the variance of the parameter distributions compared to the setup with linear groundwater inflow. Results are shown for the best 1% behavioral model parameters and reach #1 only."*

26. 415: Figure 7, it is not clear how this is different from Figure 5. New GW terminology adds confusion.

    – *We agree and will change the titles of the sub figures accordingly.*

[Figure]

*"Figure 7: Scatter plots of the best 1% behavioral model parameters ($A_{TS}$, α and the groundwater inflow qI) from calibrating the TSM with radon and chloride alone. Three*

*different model setups are presented that differ in the location of the groundwater inflow along the reach (upstream-most point, mid-point, downstream-most point model setups). For simplicity, results from calibrating the TSM with radon in the $k_{low}$ model setup are shown only."*

27. 435: "Calibrating TSMs with multiple tracers better constrains model parameters." Where are the actual results of calibrating using two tracers simultaneously? You calibrated using an individual tracer, and then calibrate to one set of combine tracer data, correct?

   - *We will explain in detail what we mean by joint and individual calibration (see response to comment no. 1).*

28. 456: "TSM is jointly calibrated with radon and chloride." Did you actually do a joint calibration? To suggest two tracers were used, a multi-objective function should be considered similar to Neilson with solute and temperature. This reviewer does not clearly follow where the joint results are.

   – *Please refer to our previous response, as well as the detailed explanation provided in response to comment no. 1.*

29. 501: "critical for contextualizing" in future radon studies? How? QLHS and Qout results look very similar to this reviewer. It just provides a longer WoD.

   – *Thank you for highlighting this point. We will revise the section and remove the term "critical for contextualizing." Furthermore, we will specify which results we explicitly refer to here to avoid confusion, clarifying that we do not refer to the $Q_{LHS}/Q_{out}$ results. The revised text will read as:*

   *"Therefore, our findings emphasize that the spatial variability of groundwater inflow location should be explicitly accounted for in future radon-based studies. This consideration may challenge the common perception that radon mass balances can be fully closed solely by including exchange with subsurface transient storage zones."*

30. 505: Statement is not clear from Figure 6. Central tendency is all the same regardless of GW location. A reach-by-reach water balance seems critical. "Location selected for GW" is arbitrary.

   – *Although the behavioral parameter distributions appear similar regardless of groundwater inflow location, our results show that these distributions differ significantly. This point will be emphasized in the main text:*

   *"We found significantly different distributions of the behavioral parameters depended on the different locations of groundwater inflow (upstream-most point, mid-point, downstream-most point model setups) independently of which tracer was used for calibration (Fig. 6, Fig. 7)."*

31. 519: "Previous studies have shown that large spatial-scale subsurface flow paths play a critical role in explaining water mass balances in streams." Redundant information, and reads too vague and may add confusion for the reader. Why are the large-spatial-scale so important? It is only important for estimating an accurate WoD. Right? Meaning, your paper is about the WoD, not regional GW processes. The point the help the reader is that local groundwater fluxes control the water balance and should be quantified as part of the in-stream BTC. That is explicit and easy to understand. Suggest to write in more explicit terms specific to your study.

- *We will clarify that "large spatial scale subsurface flow paths" do not refer to regional groundwater flow. Rather, they indicate subsurface flow paths originating further upstream in the reach. The revised text will read as:*

  *"Previous studies have shown that large spatial-scale subsurface flow paths, that are subsurface flow paths originating from further upstream of the stream reach, play a critical role in explaining water mass balances in streams (e.g., Payn et al., 2009; Stanford and Ward, 1993; Ward et al., 2023)."*

32. 540: Figure 9, should state that this figure has been modified from Payne et al. 2009. Flow path C does not seem technically correct. You do not know if that represents tracer that bypasses measurement site. The WoD is your measurement window, not the real window in which flow paths exist. And why is C red. That adds confusion. This reviewer does not find any new information added by this figure, and it does not seem technically correct.

   - *Thanks for pointing this out. We will rewrite the section with an emphasis on the novelty of labelling this flow path with radon in our study. Please see our answer to comment no. 33.*

33. 549: Just say break the transient storage zone into two zones, each with its own exchange flux. This is important for temperature and reactive solutes. It might be critical? Too vague. And critical how? How would this help with Radon? Discussion related to two-zones is a little weak; sure, denitrification representation might improve, but how is that relevant for radon and chloride?

   - *We will revise the discussion on the two-storage zone TSM and relocate it to a new section titled "Implications." The revised text will read as:*

     *"The improved parameter identifiability through joint TSM calibration with radon and chloride may be a critical step toward enhancing the physical realism of TSMs and providing more reliable estimates of solute transport. This is important because past studies found that the relationships between TSM parameters and hydrologic drivers are often contradictory (Ward and Packman, 2019; Bonanno et al., 2022). We envision that future studies will derive model parameters by jointly calibrating TSMs across diverse environments and hydrologic conditions. This approach can help clarify how model parameters vary with hydrologic drivers. To advance this goal, studies should compare streams across varying scales and geological settings to determine how these differences affect calibration outcomes when radon is included. Through such comparative studies, research could test our expectation that streams with higher radon activity, typically smaller-order streams with granite-rich geology, will yield greater certainty for groundwater inflow $q_l$ and improved model performance when TSMs are jointly calibrated with radon and chloride. This greater certainty for $q_l$ might result from the smaller greater difference in radon concentration activity between groundwater and stream water. Furthermore, future research could jointly calibrate TSMs with radon and tracer data from constant rate injections. This approach would test whether radon captures longer flow paths than those detected by constant rate injections but not by slug tracer injections.
     For future research with radon on gross exchange fluxes, we recommend constraining the transient storage parameter by considering surface (e.g., pools) and subsurface transient storage zones (e.g., hyporheic zones) in TSM evaluations (e.g., Choi et al., 2000). This distinction might be critical because only surface storage contributes to radon degassing. In contrast, radon activity increases mainly in the subsurface storage and, to a lesser extent, in surface storage. These storage-specific processes are not fully captured by using a single storage zone, as implemented in*

*OTIS-R, instead of two. We selected the one-zone storage model because this setup aligns conceptually with most previously established radon models (e.g., Cook et al., 2006; Frei and Gilfedder, 2015) and many TSM calibration with slug tracers (e.g., Bonanno et al., 2023). To our knowledge, no prior radon study has considered two storage zones, highlighting a promising opportunity for future research.*
*(…)"*

34. 563: "The goal was.."

- *We will remove the term "overarching", thanks for pointing out.*

35. 564: Radon is a solute too. Be specific here regarding which solute, which tracer.

- *We will change to:*

  *"The goal of this study was to quantify flow paths of different timescales at the reach scale using measurements of chloride concentration and naturally occurring radon."*

36. 571: Not sure this reviewer agrees that the recommendation should always be to include Radon and chloride jointly. What is missed if Radon is excluded? Can we estimate qi from groundwater first, and then use salt once that qi parameter is set? What is Radon is non-detect? Are you assuming Radon is measurable in all streams? When longer flow paths need to be identified, sure, Radon makes sense, but there is still large uncertainty in the parameter value. Plus, a one zone may be much more of a limitation than excluding Radon. Suggest to state in simple terms that Radon can extend the WoD with paired with salt, plain and simple.

- *Thank you for your suggestions. We will add a section titled "Implications and future research" to the discussion to address your recommendations. Please refer to our detailed response to comment no. 33 for more information.*

---

## Author Comment (AC2)

Please find below our responses to the comments of reviewer 2. Our responses are given in blue.

1. The authors present an interesting application of the transient storage model (TSM) in which instream tracer data is supplemented by naturally occurring radon data. The paper is generally well written and the authors have some good points. The manuscript has a number of shortcomings, however, and major revisions will be needed to develop a manuscript that warrants publication.

   The authors raise two fundamental issues in the introduction that are recurring themes in the TSM literature. I haven't done much work in this arena for some time, but I have conducted other reviews where I've made similar points. Please keep in mind that many of the comments that follow are directed at the field in general rather than these particular authors.

   The first issue is the alleged inability of the tracer-modeling approach to quantify "long flow paths". I can certainly think of cases where this is important -- e.g. when the stream crosses a fracture zone and water leaves the channel and doesn't return before the observation point (path C in Fig 9; lines 59/60) - but in general I would say that this concern is overblown. The importance of this issue is dependent on the overall goal of the study. If the goal is to quantify the processes that affect constituents that are present in the water column, such as the case of an accidental spill into a waterway, the "failure" of the approach is not of consequence -- the tracer mimics the constituent of concern, so if these long flow paths don't affect the tracer, they don't affect the constituent (the amount of mass in long flow paths is trivial). If your interest is the fate of molecules in the riparian zone as they interact with the stream (e.g. diffuse agricultural pollution that enters laterally), the long flow paths may be of greater importance. But I would argue that the typical stream tracer approach isn't an appropriate means to address this latter scenario. Further, if one is concerned about long flow paths, why use slug additions that by definition have a short window of detection? Its important to note that 100% of the data sets used to establish the transient storage paradigm were based on continuous tracer injections wherein tracer concentrations are allowed to reach a steady-state plateau. Under this approach, plateau will be achieved when all flow paths that return to the stream have had sufficient time to do so (thereby eliminating the problem associated with path B). Unfortunately many contemporary investigators have utilized slug additions as a means of streamlining the field effort, and this simplification comes with a price.

   – *We would like to thank the reviewer for these comments. We agree that, ultimately, the importance of temporally longer flow paths depends on the aims of the study. In the provided example of an accidental spill into a waterway, a slug tracer injection effectively mimics the constituent of concern. The temporally longer flow paths influence the slug tracer in the same way as they do the constituent.*
   – *Despite the well-established knowledge on the WoD inherent in slug tracer injections, we maintain that studying alternative tracers, such as radon (as proposed in our study), is critical. Alternative tracers may extend the WoD and offer additional insights into solute transport. We agree that constant-rate injections may better capture temporally longer flow paths returning to the stream. Therefore, we will propose that the joint calibration of continuous tracer injections with radon represents a promising direction for future research:*

     *"Furthermore, future research could jointly calibrate TSMs with radon and tracer data from constant rate injections. This approach would test whether radon captures temporally longer flow paths than those detected by constant rate but not by slug tracer injections."*

2. The second fundamental issue is the much celebrated parameter identifiability/equifinality problem raised by numerous investigators. The point I'd like to make here is that the "failure" to develop unique estimates of the transient storage parameters does not

necessarily indicate a "problem" with the "traditional" tracer based modeling approach. The common explanation for parameter identifiability problems is that the approach is somehow inadequate. An equally plausible explanation in many cases is that transient storage simply isn't important in the reach of interest. In the extreme case, consider data from a straight, lined canal. If you applied the approach in this situation and were unable to identify the transient storage area and exchange rate, would you blame the approach, or conclude that transient storage is unimportant? For the case of a natural stream reach, there may be identification issues simply because these flow paths aren't relevant when viewed in terms of constituent mass.

As someone who has worked for years trying to help numerous researchers analyze problematic data sets, its my opinion that the transient storage approach is adequate when applied to a quality data set. Most problems arise when the data is sparse in critical parts of the breakthrough curve, the investigators have inaccurate estimates of streamflow, and/or the data is simply noisy (due to poor lab analyses, incomplete mixing, etc).

– *Thank you for this important point. We fully agree that the "failure" of parameter identifiability could also relate to transient storage mechanisms not playing a role. In such cases, the advection-dispersion equation might be a better choice. Furthermore, we agree that the quality of the BTC could explain the lack of parameter identifiability in some cases. However, we also think that not all BTCs are simply noisy. While data quality remains a significant issue, equifinality and parameter identifiability are critical aspects extensively discussed in hydrologic research (e.g., Beven and Freer 2001).*

– *In the revised manuscript, we will incorporate the suggested alternative explanations for the "failure" of parameter identifiability:*

*"In streams where exchange between advective flow and storage zones is not minimal, non-identifiability may arise when multiple parameter combinations yield equivalent model performance."*

specific comments:
* * *
3.  1) abstract "...the amount and location of groundwater inflow, which is not explicitly accounted for in TSM".  This is incorrect, TSMs consider inflow (e.g. ql in equation 1).

    – *Thank you for this important point. We will revise the statement for improved clarity. We will change to:*

       *"This non-identifiability arises because radon activity in streams remains at steady-state and is highly sensitive to the location and amount of groundwater inflow, as well as contributions of flow paths from subsurface transient storage zones. As a result, radon measurements are biased toward longer-timescale flow paths, limiting their applicability to uniquely constrain solute transport parameters in TSM calibration without complementary slug tracers."*

4.  2) line 32  "solute tracer experiments are biased towards faster flow paths". To reiterate, if it doesn't affect the tracer, it doesn't affect the constituent of interest (at the scale studied). The tracer is simply reproducing what constituent molecules would experience.

    – *Please refer to our detailed response to comment no. 1 above.*

5.  3) line 40 "TSMs assume a uniform, steady-state ..flow".  This is incorrect -- the lateral inflow term can be used to implement non-uniform flow. Consideration of unsteady flow is rare, but possible:
    Runkel, R.L., McKnight, D.M., and Andrews, E.D., 1998. Analysis of transient storage subject to unsteady flow: Diel flow variation in an Antarctic stream, J North American Benthological Society, 17(2), 143–154, 10.2307/1467958 .

    – *Thank you for this suggestion. We will revise the text accordingly and incorporate the literature recommendation as follows:*

       *"In their simplest forms, TSMs assume a uniform, steady-state, one-dimensional flow, modeled using the advection-dispersion equation (ADE), while also accounting for first-order mass transfer between the advective flow and a storage zone (Bencala and Walters, 1983; Gooseff et al., 2008). Extensions allow for implementing non-uniform groundwater inflow via lateral inflow terms (Runkel et al., 1998)."*

    also, I don't agree with "effectively infinite dimensions" (nor does your text on line 154, "finite-size, well-mixed storage zone")

    – *We will remove this aspect from the text; please refer to our response to the previous comment for the improved text.*

6.  4) line 45, "The parameter values derived from TSMs provide a means of comparing solute transport within a single stream or across multiple streams". Could cite:
       Runkel, R.L., 2002. A new metric for determining the importance of transient storage, J. North American Benthological Society, 21(4), 529–543, 10.2307/1468428 .

- *We will follow the suggestion.*

7. 5) lines 50-55. Again, it depends on your objective; for most cases I'm not convinced adding the radon data really helps. I find the idea of supplementing the tracer data with other auxiliary data the most promising approach for separating surface and subsurface (hyporheic) storage. I think you allude to using radon data for this purpose later in the paper. For more of my thoughts (as if you haven't had enough :-), see:

   > Runkel R.L., McKnight, D.M., and Rajaram, H., 2003. Modeling hyporheic zone processes, Advances in Water Resources, 26(9), 901–905, 10.1016/S0309-1708(03)00079-4

   – *Thanks for pointing this out. We will adapt parts of the discussion to highlight the research opportunity of using radon in a two-storage zone TSM.*

   *"We selected the one-zone storage model because this setup aligns conceptually with most previously established radon models (e.g., Cook et al., 2006; Frei and Gilfedder, 2015) and many TSM calibration with slug tracers (e.g., Bonanno et al., 2023). To our knowledge, no prior radon study has considered two storage zones, highlighting a promising opportunity for future research."*

8. 6) line 79-80. "When surface water exchanges with subsurface transient storage zones and contacts radium-bearing minerals in the streambed, radon activity increases as a function of the time spent in the hyporheic zone". Radon in the stream could come from this contact w/ streambed materials AND/OR groundwater inflow. I'm guessing the streambed part is modeled using the production term (gamma, equation 2) and the groundwater part is handled through the lateral inflow term. More description of how this is handled should be added, including what you're using to set the lateral inflow concentration.

   – *We will follow the suggestion The revised version will read as:*

   *"We calculated the radon production term γ as the product of the decay constant ($0.18$ $d^{-1}$) and the measured equilibrium radon activity (Gilfedder et al. 2019). This approach assumes that radon production occurs only in the subsurface transient storage zone of the stream. However, radon may also increase when stream water interacts directly with the streambed surface."*

   *"For all three calibration approaches, the measured equilibrium radon activity was used as the activity of the groundwater inflow."*

9. 7) line 108-110. "Each reach length was at least 20 times the Wetted Channel Width to control for expected variations in solute transport that occur as a function of reach selection". Similar metrics for reach length are often mentioned in regard to complete mixing of the tracer with depth and width, but I'm guessing you're alluding to the Dahmkohler number. You may want to clarify this.

   – *We do not explicitly refer to the Damköhler number in our study; however, we will revise the sentence to clarify the rationale behind the chosen reach length:*

   *"Each reach length was at least 20 times the Wetted Channel Width. This was done to ensure thorough mixing of the tracer and solutes across both the width and depth of the channel, thereby minimizing spatial variability and potential biases due to reach selection"*

10. 8) line 120. "Discharge was calculated for the resulting BTCs using dilution gaging". I assume this required some relationship between conductivity and chloride, which is fine. But you may want to try this method as a check:

McCleskey, R. B., Runkel, R. L., Murphy, S. F., & Roth, D. A. (2025). Stream discharge determinations using slug additions and specific conductance. Water Resources Research, 61, e2024WR037771. https://doi.org/10.1029/2024WR037771 .

I'm happy to help and provide an updated spreadsheet if interested.

– *Thank you for highlighting this interesting manuscript. We agree that the method represents an important contribution to discharge estimation and will consider this approach in future research.*

11. 9) line 128. "Radon sampling sites were co-located with BTCs observations" - was there 1 sample per site? This paper is all about the radon yet we don't get to see the data - please add.

– *We present the radon data in the results section, but only report the minimum and maximum values for the stream. A detailed dataset will be provided in the appendix.*

12. 10) line 187-188. Here you refer to D, alpha, Ats; Table 1 has D, A, Ats

– *Thank you for this helpful suggestion. We will revise the table accordingly to improve clarity and accuracy.*

13. 11) line 188-189. Is it a uniform distribution or a logarithmic one?, I'm confused...

– *We performed uniform sampling in logarithmic space. To clarify this, we will rewrite the sentence as follows:*

*"We sampled parameter values of $D$, $\alpha$, and $A_{TS}$ uniformly from a log10 transformed space to ensure approximately equal representation for each order of magnitude within the parameter space (Kelleher et al. 2013; Ward et al. 2017)."*

14. 12) line 195, "intersection of behavioral parameter sets" - I like this approach...

– *Thank you!*

15. 13) calibration approach described in section 2.4 and various flow approaches described later in the paper. I don't agree with the approach of fixing velocity and A, and estimating Q for several reasons. I highly recommend fixing Q and estimating A. A few subpoints:
- why not use the Q from the slug additions? I think your three Q methods overly complicate things and these complications aren't relevant to what you're trying to show (the utility of adding radon data). I suggest using the Qfix approach and dropping the others. If you're worried about uncertainty in your slug estimates, develop a linear regression between the Q estimates and distance and use the regressed values at each site.

– *The reason we apply three different calibration approaches is not to demonstrate the utility of adding radon. Instead, we aim to show why only using radon does not work.*

- *Only by using different calibration approaches can we demonstrate that:*
  - o *The radon activity in the stream is highly sensitive to inflowing groundwater; even within the measurement uncertainty of discharge calculated from slug tracers, we observe differences in model performance ($Q_{fix}$).*
  - o *A higher degree of freedom in the calibration method ($Q_{LHS}$) compared to $Q_{fix}$ improves model performance, as indicated by a greater number of consistent behavioral parameter sets for both radon and chloride.*
  - o *Furthermore, calibrating groundwater inflow allows us to infer water loss ($Q_{out}$) with radon calibration, an insight we do not obtain by calibrating the TSM to slug injections.*

- *We are not concerned about uncertainty in slug tracer estimates. Rather, our effort to apply three calibration approaches highlights the uncertainty inherent in radon. We will clarify this aspect in the manuscript:*

  *"Radon activity in streams varies with the amount of inflowing groundwater, as radon activity differs significantly between groundwater and surface water (Cook, 2013). Small changes in the amount of inflowing groundwater may lead to differences in model performance. To account for this, we either calibrated or calculated groundwater inflow within three different calibration approaches, in addition to calibrating D, α, and $A_{TS}$."*

- fixing A does nothing to reduce "potential issues of equifinality" (line 205). The main channel area is by far the easiest parameter to estimate via simulation as it controls the velocity and thus the timing of the BTC. When estimating A, Ats, alpha and D using nonlinear regression, the A parameter always has the narrowest 95% confidence interval and is the parameter estimated with the most certainty. Equifinality problems usually arise when there's not enough data in certain parts of the BTC to uniquely identify D, Ats, and alpha. Wagner and/or Harvey have papers (book chapters?) which show the sensitivity of various parts of the BTC for various parameters and there's always ample info to estimate A. By fixing A (and/or velocity) you're ignoring this information and biasing your other parameter estimates. If you insist on fixing A/velocity, I suggest using the center of mass rather than the peak as this more truly represents the average reach velocity (see Runkel 2002 ref above), especially if there's an extended tail.

- *Thank you for mentioning this important point. We agree that different parts of the BTC provide distinct information on specific parameters, as demonstrated by previous studies using slug and constant-rate injections (e.g., Wagener et al. 2002; Wlostowski et al. 2013; Kelleher et al. 2013; Wagner and Harvey 1997). However, research does not consistently identify which part of the BTC corresponds to which information (see Figure 8 in Bonanno et al. (2022) for a visualization of these differences). Recently, Bonanno et al. (2022) systematically studied the effect of calibrating stream velocity on parameter identifiability in BTCs. They concluded that fixing the stream velocity is reasonable and does not affect the identifiability of the remaining parameters. We followed this approach since, to our knowledge, it is the most recent and state-of-the-art method, and the only systematic study that explicitly studied the calibration of stream velocity. Therefore, we consider fixing A/velocity to be reasonable for our study. A detailed explanation of this aspect will be provided in the manuscript:*

  *"This choice was motivated by findings from Bonanno et al. (2022), who showed that $A_{TS}$ and α are often not identifiable when v is calibrated instead of calculating v by dividing the stream length by the arrival time of the concentration peak of the downstream BTC."*

16. 14) lines 215-220. Degassing is a function of turbulence which is effected by velocity and thus Q. Stream width (surface area) is also important. Why not use the value estimated at the most similar Q?

– *We agree that the discharge at the most similar Q condition is likely to best reflect the experimental situation. However, Cargill et al. (2011) did not find a clear functional relationship (linear, exponential, or otherwise) between Q and k and investigated only three Q conditions. We therefore consider Cargill et al.'s (2011) results to be a reasonable approximation, but not sufficiently certain to be fixed at a single value. In addition, we explicitly test whether a potential misestimation of k would alter the conclusions. This ensures that our findings are not an artifact of choosing one particular k value, but remain valid across a plausible range.*

17. 15) line 334. "radon provided more information on groundwater inflow". With all the uncertainty involved (e.g. degassing rate, radon inflow concentration, radon analyses) why would these estimates of inflow be better than your dilution gaging? The Qfix approach is the way to go.

– *We do not claim that radon provides more information on groundwater inflow than derived from dilution gauging. However, our quantitative results on information content show that TSM calibrations with radon offer more information than calibration with chloride concentration on groundwater inflow, which is not related to dilution gauging. This is likely because radon activity increases with inflowing groundwater, whereas chloride concentrations are diluted. We will clarify this aspect in the revised version of the manuscript:*

*"Although we expect groundwater inflow to primarily affect radon activity in streams, we also calibrated TSM parameters in three model setups that varied in groundwater locations using chloride concentrations. This was motivated by the assumption that chloride-free groundwater, as commonly assumed in TSMs, dilutes chloride concentrations in streams."*

18. 16) line 402. "parameter interactions became evident when inflow was at the most downstream points". Back on line 310 you mention all of the inflow entering over 1 meter long area; it's my experience that these abrupt changes can cause numerical problems -- you may not 'see' these problems at sites farther away from the observation point (the Crank-Nicolson method usually 'recovers'), but this one at the downstream end could be a numerical artifact. I suggest spreading the inflow over several 1-m segments (maybe 10 m).

– *Thank you for raising this important concern. As part of our calibration routine, we always check the numerical stability of the model run as a prerequisite to the actual calibration (see our publicly available calibration code in Bacher et al., 2025). All presented results therefore stem from numerically stable calibrations. Therefore, we can confirm that the calibration with groundwater inflow at the most downstream location is not a numerical artefact and we are confident that the numerical stability we observe is unrelated to any 'recovery' of the Crank–Nicolson method. The same data were used in all three setups, with only the groundwater inflow location varied.*
– *However, we speculate that numerical instabilities may become more significant when higher lateral groundwater inflow is calculated or calibrated. In our simulations with three different inflow locations, we used data from reach #1 for calibration and discharge was calibrated within a parameter range defined by BTC measurements. The discharge in reach #1 did not change significantly (see Figure 8), which led to relatively low inflow values and may explain the numerical stability.*

19. 17) line 470. "This suggests that obtaining narrow, well-constrained estimates for groundwater inflow, ATS, or α from calibrating the TSM with radon will remain challenging unless at least one of these parameters is further constrained" -- this is easily done - fix Q and save the modeling approach for the more empirical/abstract parameters!!!

– *Please refer to our response to Comment No. 15 for a detailed reply.*

Beven, Keith; Freer, Jim (2001): Equifinality, data assimilation, and uncertainty estimation in mechanistic modelling of complex environmental systems using the GLUE methodology. Journal of Hydrology 249 (1-4), pp. 11–29. DOI: 10.1016/S0022-1694(01)00421-8.

Bonanno, Enrico; Blöschl, Günter; Klaus, Julian (2022): Exploring tracer information in a small stream to improve parameter identifiability and enhance the process interpretation in transient storage models. Hydrol. Earth Syst. Sci. 26 (23), pp. 6003–6028. DOI: 10.5194/hess-26-6003-2022.

Kelleher, C.; Wagener, T.; McGlynn, B.; Ward, A. S.; Gooseff, M. N.; Payn, R. A. (2013): Identifiability of transient storage model parameters along a mountain stream. Water Resour. Res. 49 (9), pp. 5290–5306. DOI: 10.1002/wrcr.20413.

Rodriguez, Nicolas Björn; Pfister, Laurent; Zehe, Erwin; Klaus, Julian (2021): A comparison of catchment travel times and storage deduced from deuterium and tritium tracers using StorAge Selection functions. Hydrol. Earth Syst. Sci. 25 (1), pp. 401–428. DOI: 10.5194/hess-25-401-2021.

Wagener, T.; McIntyre, N.; Lees, M. J.; Wheater, H. S.; Gupta, H. V. (2003): Towards reduced uncertainty in conceptual rainfall-runoff modelling: dynamic identifiability analysis. Hydrol. Process. 17 (2), pp. 455–476. DOI: 10.1002/hyp.1135.

Wagener, Thorsten; Camacho, Luis A.; Wheater, Howard S. (2002): Dynamic identifiability analysis of the transient storage model for solute transport in rivers. Journal of Hydroinformatics 4 (3), pp. 199–211. DOI: 10.2166/hydro.2002.0019.

Wagner, Brian J.; Harvey, Judson W. (1997): Experimental design for estimating parameters of rate-limited mass transfer: Analysis of stream tracer studies. Water Resour. Res. 33 (7), pp. 1731–1741. DOI: 10.1029/97WR01067.

Wlostowski, Adam N.; Gooseff, Michael N.; Wagener, Thorsten (2013): Influence of constant rate versus slug injection experiment type on parameter identifiability in a 1-D transient storage model for stream solute transport. Water Resour. Res. 49 (2), pp. 1184–1188. DOI: 10.1002/wrcr.20103.